# In-utero exposure to chikungunya and child morbimortality: a population-based study using linked routine data

Mio Kushibuchi [1] ✉, Orlagh Carroll[2], Thiago Cerqueira-Silva [2,3], Viviane S. Boaventura [3,4], Maria Glória Teixeira[5], Mauricio L. Barreto[1] & Enny S. Paixão [1,2]

Chikungunya exposure in-utero is linked to neonatal morbidity and neuro-developmental effects. We examined the long-term morbidity associated with in-utero Chikungunya. This registry-based cohort study linked records of infants born in Brazil between 2015 and 2018, with all-cause first hospitalization and death as outcome. Infants were followed until the outcome, their third birthday, or the end of the study. Adjusted stratified Cox models were used to estimate hazard ratios (HR), 95% confidence intervals (95% CIs), and absolute risk differences. A total of 1,821 exposed and 18,210 unexposed infants were included. The HR for hospitalization was 1.21 (95% CI: 1.11–1.36), corresponding to 37 excess hospitalizations per 1000 exposed (95% CI: 16-64). The risk was twofold for intrapartum exposure (HR 2.08, 95% CI: 1.33–3.44) and elevated for first- and second-trimester exposure. Evidence for risk of death was limited. Here we show an elevated hospitalization risk associated with in-utero Chikungunya exposure.

Chikungunya fever (CHIKF), a mosquito-borne Arbovirus infection, has spread globally and is now recognized as an emerging global health threat associated with climate change[1]. In 2024, over 600,000 cases of CHIKF were reported worldwide[2]. Climate change and increasing global mobility have expanded the range of affected regions, with autochthonous CHIKF cases detected in France and Italy, as well as imported cases among travelers in the United Kingdom in 2025[2,3]. Brazil accounts for approximately 97% of global CHIKF cases[2], with considerable regional and socioeconomic disparities in disease burden. Increasing evidence indicates that CHIKF can lead to long-term health consequences, including chronic chikungunya and elevated mortality among affected individuals even after the acute period[4].

Mother-to-child transmission of the CHIK virus was first documented during the La Réunion outbreak[5]. Among the in-utero exposed neonates (i.e., the neonates whose mothers had CHIKF during pregnancy), vertical transmission was exclusively observed in infection near delivery, with 50% of the infants born to mothers with intrapartum viremia having a vertical transmission[5]. Severe neonatal morbidity and long-term sequelae have been documented in cases of vertical transmission, exhibiting sepsis-like symptoms of fever and irritability[6]. Systematic studies also show similar vertical transmission rates for intrapartum infection, with neonatal death occurring in 2.8% of neonates with vertical transmission[6,7].

The consequences of in utero chikungunya virus infection occurring at earlier gestational ages have been examined in several studies, but findings remain inconsistent. A study from La Réunion comparing birth outcomes between infected and uninfected pregnant women found no differences in neonatal hospitalization, preterm birth, birth weight, congenital anomalies, or stillbirth[8]. Similarly, a study from Grenada[9] and a cross-sectional study from Thailand[10] reported no increase in neonatal morbidity or birth complications. In contrast, a case–control study from French Guiana[11] identified a higher

[1]Centre for Data and Knowledge Integration for Health (CIDACS), Fundação Oswaldo Cruz, Salvador, Brazil. [2]Faculty of Epidemiology and Population Health, London School of Hygiene and Tropical Medicine, London, UK. [3]The Gonçalo Moniz Institute (IGM), Fundação Oswaldo Cruz, Salvador, Brazil. [4]Faculty of Medicine of Bahia, Universidade Federal da Bahia, Salvador, Brazil. [5]Institute of Collective Health (ISC), Federal University of Bahia, Salvador, Brazil. ✉e-mail: Miokushibuchi0119@gmail.com

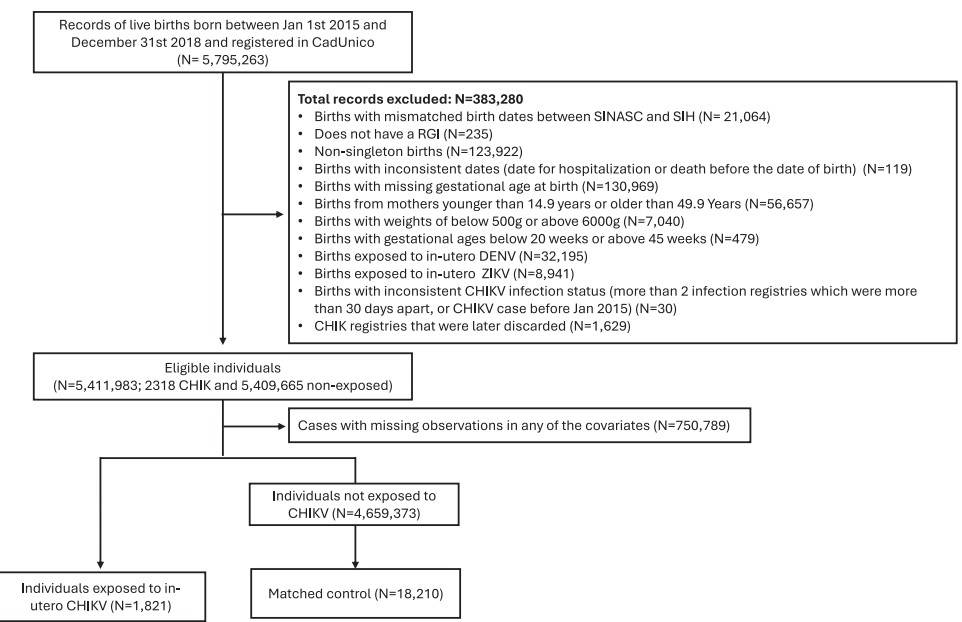

**Fig. 1 | Flowchart for the exclusion and matching of participants.** CadUnico, Unified Registry for Social Programs (Cadastro Único para Programas Sociais); CHIKV, Chikungunya virus; DENV, Dengue virus; RGI, Immediate Geographical Region (Região Geográfica Imediata); SIH, Hospital Information System (Sistema de Informações Hospitalar); SINASC, Live Birth Information System (Sistema de Informações Sobre Nascidos Vivos); ZIKV, Zika Virus.

rate of neonatal ICU admissions among CHIKF-exposed newborns, and a cohort study from Nigeria reported congenital anomalies, including cleft lip/palate and microcephaly[12]. A systematic review shows the antepartum fetal death risk of 1.7%[6]. More recently, our team, using the CIDACS Birth Cohort in Brazil, observed increased risks of preterm birth, low Apgar scores, and neonatal death among neonates exposed to CHIKF in utero[13].

The literature on long-term outcomes following in utero exposure to Chikungunya remains limited, with most studies focusing on neurodevelopmental effects. Studies from Grenada[14] and Brazil[15], which primarily assessed antenatal maternal infection without confirmed fetal or neonatal infection, report no increased risk of neurological sequelae. Novel evidence from La Réunion documented cognitive and learning difficulties at age 10 in 57.9% of exposed children, with the risk nearly doubled among those who experienced neonatal encephalopathy[16]. In contrast, non-neurocognitive long-term outcomes, such as general morbidity requiring hospitalization, are notably under-investigated. Given the substantial burden of CHIKF, its acute and chronic health effects, and the potential for severe disease among vertically infected neonates, we hypothesized that in utero CHIKF exposure may increase the risk of hospitalization and mortality beyond the neonatal period.

This study investigates long-term morbidity and mortality associated with in utero exposure to CHIKF by comparing all-cause hospitalization and mortality during the first three years of life between exposed and unexposed infants. We also assess whether risks differ by gestational age at infection. We focus on all-cause hospitalization and mortality as broad indicators of early-life health impacts.

## Results
### Study population and baseline characteristics
Between 2015 and 2018, a total of 5,795,265 live births were recorded in the CIDACS Birth Cohort. After applying the exclusion criteria, 1821 CHIKF-exposed and 18,210 matched unexposed live births were included in the analysis (Fig. 1). The characteristics of the unexposed infants that were matched and included in the study were comparable to those of infants who were not matched (Supplementary Table 1).

Table 1 shows the basic demographics, overall and by exposure status. The exposed group had a higher percentage of individuals residing in urban areas (83% versus 72%, SMD = 0.253) and who received adequate antenatal care (83% versus 77%, SMD = 0.136). The distribution of year of conception was also different between the groups, with the exposed group concentrated in 2015 and 2016 (SMD = 0.774). Regarding the observed outcomes, a higher percentage of individuals in the exposed group experienced hospitalization: 393 out of 1821 (21.6%) were hospitalized, compared to 3242 out of 18,210 (17.8%) in the non-exposed group. A total of 215 deaths (1.1%) occurred, comprising 21 (1.2%) in the exposed group and 194 (1.1%) in the non-exposed group (Supplementary Tables 2, 3).

### All-cause first hospitalization up to age three
A Kaplan-Meier survival curve for all-cause first hospitalization is shown in Fig. 2. The median follow-up time was 18.7 months [IQR: 28.1], with no difference between the exposed and non-exposed. Over a three-year follow-up, those exposed to CHIKF during pregnancy had an adjusted hazard of hospitalization 1.21 times that of the non-exposed (95% CI: 1.11–1.32) (Fig. 3, Supplementary Table 4). In an absolute scale, in-utero CHIKF exposure was associated with 37.3 additional hospitalizations per 1000 exposed infants over the first three years of life (95% CI: 16.8–64.2) (Fig. 4, Supplementary Table 5). The Cox models met the proportional hazards assumption (Supplementary Table 6). In the analyses investigating the trimester of maternal infection, the HR for exposure during the first trimester was 1.35 (95% CI: 1.10–1.67), and for second trimester, 1.25 (95% CI: 1.06–1.48); whereas for the third trimester the results showed no evidence of an association (HR:1.06, 95% CI: 0.86–1.30). The risk of hospitalization was markedly elevated among infants exposed in the intrapartum period, with an HR of 2.08 (95% CI: 1.33–3.44) compared with non-exposed infants (Fig. 3, Supplementary Table 4). The association between CHIKF during pregnancy and neonatal hospitalization was not statistically significant (HR: 1.07, 95% CI: 0.93–1.26) (Supplementary Table 7).

Among the 356 hospitalizations in the exposed group, 44.3% were due to perinatal causes, 15.8% due to infection, and 19.8% due to

**Table 1 | Basic demographics overall and by CHIKF exposure status (*N* = 20,031)**

| Overall (*N* = 20,031) | | CHIKF exposure status | | SMD |
|---|---|---|---|---|
| | | Non-exposed (*N* = 18,210) | Exposed (*N* = 1821) | |
| **Sex** | | | | |
| Female | 9741 (48.6) | 8849 (48.6) | 892 (49.0) | 0.026 |
| Male | 10287 (51.4) | 9359 (51.4) | 928 (51.0) | |
| missing | 3 (0.0) | 2 (0.0) | 1 (0.1) | |
| **Birth weight (grams)** | | | | |
| mean (SD) | 3249.46 (534.66) | 3248.86 (534.10) | 3255.45 (540.35) | 0.012 |
| 501–1000 g | 77 (0.4) | 71 (0.4) | 6 (0.3) | 0.026 |
| 1001–1500 g | 109 (0.5) | 98 (0.5) | 11 (0.6) | |
| 1501–2500 g | 1160 (5.8) | 1058 (5.8) | 102 (5.6) | |
| 2501–4000 g | 17447 (87.1) | 15863 (87.1) | 1584 (87.0) | |
| 4001–6000 g | 1236 (6.2) | 1118 (6.1) | 118 (6.5) | |
| missing | 2 (0.0) | 2 (0.0) | 0 (0.0) | |
| **Gestational age at birth, weeks** | | | | |
| mean (SD) | 38.72 (2.22) | 38.71 (2.23) | 38.83 (2.14) | 0.056 |
| 20–27 | 89 (0.4) | 82 (0.5) | 7 (0.4) | 0.06 |
| 28–31 | 180 (0.9) | 167 (0.9) | 13 (0.7) | |
| 32–36 | 1800 (9.0) | 1661 (9.1) | 139 (7.6) | |
| 37–46 | 17962 (89.7) | 16300 (89.5) | 1662 (91.3) | |
| **Marital status** | | | | |
| Divorced | 137 (0.7) | 125 (0.7) | 12 (0.7) | 0.058 |
| Married | 4313 (21.5) | 3953 (21.7) | 360 (19.8) | |
| Single | 9499 (47.4) | 8599 (47.2) | 900 (49.4) | |
| Stable union | 5916 (29.5) | 5385 (29.6) | 531 (29.2) | |
| Widowed | 38 (0.2) | 33 (0.2) | 5 (0.3) | |
| missing | 128 (0.6) | 115 (0.6) | 13 (0.7) | |
| **Maternal education, years** | | | | |
| No formal education | 126 (0.6) | 116 (0.6) | 10 (0.5) | |
| 1–3 years | 744 (3.7) | 673 (3.7) | 71 (3.9) | 0.101 |
| 4–7 years | 4666 (23.3) | 4276 (23.5) | 390 (21.4) | |
| 8–11 years | 12969 (64.7) | 11725 (64.4) | 1244 (68.3) | |
| 12 or more years | 1526 (7.6) | 1420 (7.8) | 106 (5.8) | |
| **Maternal race/ethnicity** | | | | |
| Asian | 42 (0.2) | 40 (0.2) | 2 (0.1) | 0.078 |
| Black (Preto) | 867 (4.3) | 799 (4.4) | 68 (3.7) | |
| Indigenous | 114 (0.6) | 108 (0.6) | 6 (0.3) | |
| Mixed (Pardo) | 16827 (84.0) | 15256 (83.8) | 1571 (86.3) | |
| White | 2181 (10.9) | 2007 (11.0) | 174 (9.6) | |
| **Maternal age, years** | | | | |
| mean (SD) | 25.42 (6.43) | 25.38 (6.42) | 25.90 (6.56) | 0.081 |
| 15 to 19 | 4092 (20.4) | 3747 (20.6) | 345 (18.9) | 0.041 |
| 20 to 34 | 13866 (69.2) | 12583 (69.1) | 1283 (70.5) | |
| 35 to 49 | 2073 (10.3) | 1880 (10.3) | 193 (10.6) | |
| **Number of ANC** | | | | |
| Adequate | 15648 (78.1) | 14137 (77.6) | 1511 (83.0) | |
| One time less | 1811 (9.0) | 1678 (9.2) | 133 (7.3) | |
| Two or more times less | 2572 (12.8) | 2395 (13.2) | 177 (9.7) | 0.136 |
| **Previous pregnancies** | | | | |
| None | 7002 (35.0) | 6384 (35.1) | 618 (33.9) | 0.024 |
| One or more | 13029 (65.0) | 11826 (64.9) | 1203 (66.1) | |
| **Urbanicity of residence** | | | | |
| Rural | 5313 (26.5) | 5003 (27.5) | 310 (17.0) | 0.253 |
| Urban | 14718 (73.5) | 13207 (72.5) | 1511 (83.0) | |

**Table 1 (continued) | Basic demographics overall and by CHIKF exposure status (N = 20,031)**

| Overall (*N* = 20,031) | | CHIKF exposure status | | SMD |
|---|---|---|---|---|
| | | Non-exposed (*N* = 18,210) | Exposed (*N* = 1821) | |
| Year of conception | | | | |
| 2014 | 4270 (21.3) | 4252 (23.3) | 18 (1.0) | 0.774 |
| 2015 | 5050 (25.2) | 4396 (24.1) | 654 (35.9) | |
| 2016 | 4729 (23.6) | 4116 (22.6) | 613 (33.7) | |
| 2017 | 4731 (23.6) | 4259 (23.4) | 472 (25.9) | |
| 2018 | 1251 (6.2) | 1187 (6.5) | 64 (3.5) | |
| All-cause hospitalization up to age 3 | 3635 (18.1) | 3242 (17.8) | 393 (21.6) | 0.095 |
| All-cause death up to age 3 | 215 (1.1) | 194 (1.1) | 21 (1.2) | 0.008 |

*ANC* antenatal care; *APGAR* Appearance, pulse, grimace activity, and respiration score; *CHIKF* Chikungunya virus, *SMD* standard mean difference.

respiratory causes. The distribution was similar in the non-exposed group (Supplementary Table 8, Fig. 1).

### All-cause death up to age three

Over the three-year follow-up period, the median follow-up time was 25.3 months [IQR: 23.4], with no difference between the groups. There was no difference in the hazard of death between exposed and non-exposed to CHIKF (HR: 1.01, 95% CI: 0.58–1.62) (Fig. 3, Supplementary Table 4). HR for intrapartum exposure may be elevated, but due to the small number of events, the estimate was imprecise (HR: 6.17, 95% CI: 0.94–34.9). Analyses stratified by the trimester of exposure and for neonatal death showed no evidence of an association. (Supplementary Tables 4, 7).

### Sensitivity analyses

The results and the number of observations for each sensitivity analysis are shown in Supplementary Tables 9, 10. The first sensitivity analysis, restricted to normal birthweight and term births (*N* = 15,544), replicated the main analyses' results for hospitalization (HR: 1.24, 95% CI: 1.09–1.40) and death (HR: 0.80, 95% CI: 0.36–1.75). The second sensitivity analysis, restricted to laboratory-confirmed exposed cases (*N* = 699 exposed and *N* = 6990 matched controls), was not statistically significant for hospitalization (HR:1.02, 95% CI: 0.85–1.23) nor death (HR: 0.96, 95% CI: 0.41–1.33). The results using the intrapartum definition of two days before to two days after birth, aligning with the definition used in CHMIERE cohort[5], yielded 29 individuals in intrapartum strata. The HR for admission was somewhat larger than the main analyses (HR: 4.25, 95% CI: 2.14–8.44), while the death hazards were not significant (HR: 2.72, 95% CI: 0.45–16.48). The analyses stratified by sex yielded similar HRs for admission for both sexes (HR:1.32, 95% CI: 1.12–1.56 for males; HR: 1.21, 95% CI: 1.01–1.46 for females).

## Discussion

In this registry-based longitudinal study, we found that in utero exposure to CHIKF was associated with an HR for hospitalization of 1.21, corresponding to 37 excess hospitalizations per 1000 exposed infants. The risk was particularly elevated for intrapartum exposure, with a twofold increased risk. Elevated risks were also observed for exposures during the first and second trimesters. The leading causes of hospitalization were similar for the exposed and unexposed groups. Analyses of death hazards were overall inconclusive due to limited events, although they suggest a possible elevation in risk, especially for intrapartum exposure.

Our study adds a unique finding to the body of evidence on the long-term effects of in-utero CHIKF infection. Adverse effects of intrapartum CHIKF exposure have been previously described[7], and consistent with these findings, our study found that intrapartum exposure was associated with double the risk of hospitalizations and potentially of mortality. The evidence of the harmful effects of early

gestation exposure to CHIKF is limited, as most population studies have not investigated the effect by gestational age. A previous study from our group has shown an increase in neonatal mortality among those exposed in the second trimester[13]. We could not reproduce the findings of this prior research due to an insufficient sample size, as we restricted the analysis to infants registered in CADU to avoid misclassifying hospitalizations among those with private health insurance.

We hypothesize several biological mechanisms that can explain our findings. First, maternal immune activation (MIA) could be altering the fetus's immune system, as a study showed elevated cytokines in blood samples from newborns exposed to intrapartum CHIKF[17]. MIA could cause epigenetic changes in the fetus, leading to increased long-term morbidity[18]. The health status at birth, such as intrauterine growth restrictions and neonatal complications, may also affect the long-term morbidity. The CHIKF virus may infect the placenta, which can cause fetal distress. A pathological study of placental tissue from miscarriages following early pregnancy CHIKF reported signs of placental infection[19]. Maternal CHIKF infection could be exacerbating preexisting pregnancy complications, for studies have shown that CHIKF infection exacerbates preexisting comorbidities[20]. Furthermore, CHIKF causes chronic joint pain, and pregnant women could be taking non-prescribed analgesics. Use of over-the-counter medication is common in Brazil[21], including non-steroidal anti-inflammatory drugs (NSAIDS), which may cause fetal adverse events[22]. The placental infection, exacerbated pregnancy complications, and use of analgesics can compromise the neonate's health, leading to elevated morbidity during infancy. Regarding the elevated morbidity associated with intrapartum exposure, clinical studies from La Réunion have shown that the mechanisms of vertical transmission are direct contact between the fetal circulation and maternal blood with a high viral load, and placental micro-lesions in the placental membranes during uterine contractions[5]. Our study has limited clinical data, and further studies are needed to determine whether the elevated HR observed for intrapartum exposure was due to vertical transmission.

This study has several strengths. First, it has a large sample size of over 1800 exposed cases and a long follow-up period of up to three years. Another strength is our robust study design. We chose a matched cohort to minimize exposure misclassification by ensuring that both exposed and unexposed individuals were drawn from the same regions and time periods and were thus subjected to the same surveillance, health care access and reporting practices. This design also accounted for the sinusoidal seasonal pattern of CHIKF[23] and the heterogeneity in incidence across regions[4].

There are several limitations. Because early fetal losses could not be included, the findings are subject to live birth bias, potentially underestimating the CHIKF burden. CHIKF infection in early pregnancy can lead to miscarriage or stillbirths[24], as shown by systematic reviews[6] and pathological studies[21]. These events, especially early miscarriages, are likely underreported in resource-limited settings[25].

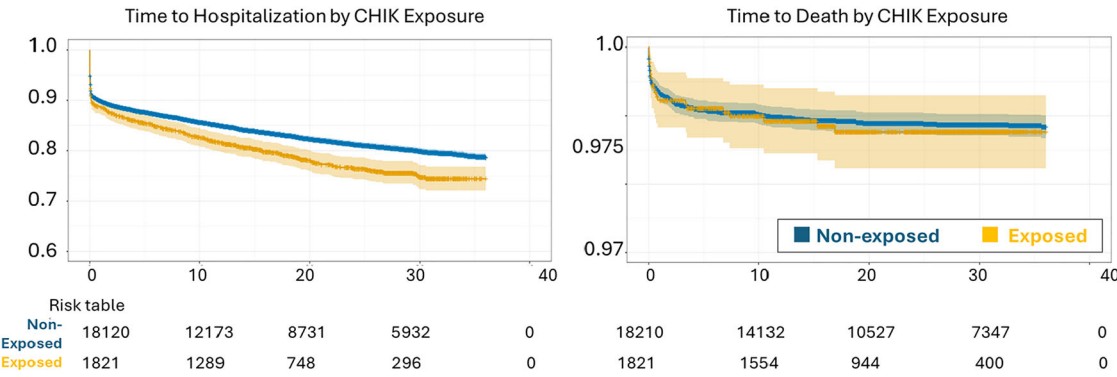

**Fig. 2 | Kaplan-Meier curve and risk table for time to first hospitalization and death (N = 34,694).** CHIK, Chikungunya.

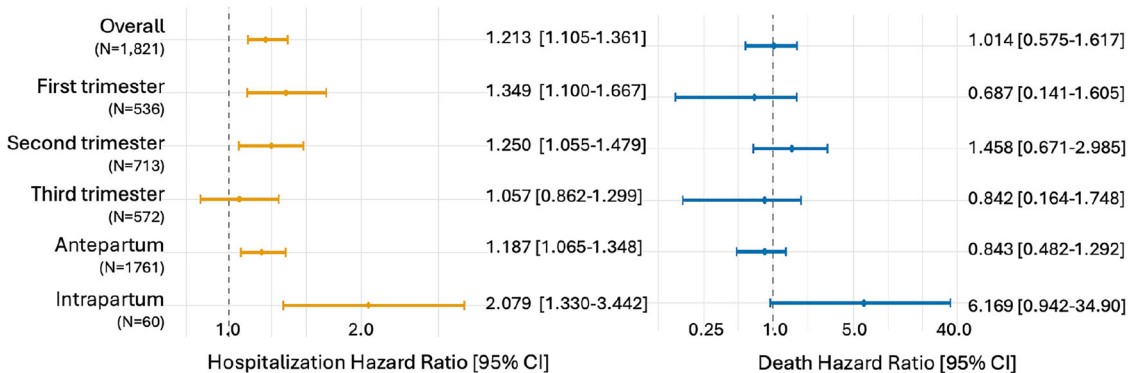

**Fig. 3 | Risks of hospitalization in the first three years of life by CHIKF exposure, described with hazard ratio (HR) and the 95% confidence interval (CI).** The numbers displayed in brackets for each stratum are the exposed, and the HR is obtained compared with the unexposed (N = 31,540). HR is adjusted for maternal age, education, race/ethnicity, and previous gestations, residential urbanicity, number of antenatal care visits, and year of conception. (no footnote for Fig. 3).

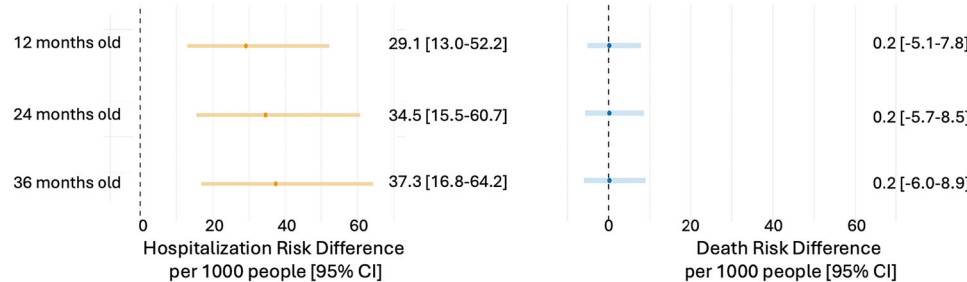

**Fig. 4 | Risk difference for hospitalization and death by CHIKF exposure, calculated for the marginal effect at age points 12 months, 24 months and 36 months (N = 34,694).** Cox model adjusted for maternal age, education, race/ethnicity, and previous gestations, residential urbanicity, number of antenatal care visits, and year of conception was used to compute the marginal risks for counterfactual populations. (no footnote for Fig. 4).

Secondly, our study relies on passive surveillance of patients who presented to a health center. Pregnant women with asymptomatic CHIKF, or those facing barriers to accessing healthcare, may not have been captured, leading to potential exposure misclassification. Such misclassification would most likely involve exposed women being misclassified as unexposed, biasing our estimates toward the null. We also observed that exposed individuals were more likely to have received adequate antenatal care, likely reflecting their greater concentration in urban areas where healthcare access is generally better. To address these, we adjusted for both antenatal care adequacy and residential location in our models. Misclassification of outcomes is also possible. Although deaths are unlikely to go unreported, the SIH hospitalization records capture only admissions to public hospitals in Brazil. Because our study population consists of individuals registered in CADU, it is unlikely that they would seek care in private facilities; however, the few cases hospitalized in private hospitals or those who migrated out of the country would not be captured in our data. Third, we lacked data on key confounders such as family income and household living conditions. These factors may contribute to residual confounding that cannot be fully eliminated. Although our study design and the covariates included in the models may partially account for these, they do not completely address them. We also did not have information on pregnancy complications that could modify the effect of CHIKF on offspring morbidity and mortality. Fourth, we performed a complete-case analysis. About 14% of the eligible population was excluded due to missing a covariate, and if missingness was associated with the outcome, the results may be biased. Fifth, indication bias may have influenced hospitalization patterns among infants exposed in the

intrapartum period. The risks associated with intrapartum CHIKF exposure are well recognized in the literature, and they could prompt clinicians to hospitalize the newborns for closer observation even in the absence of symptoms. Such precautionary admissions, if occurring shortly after birth, may artificially shorten the time to first hospitalization and inflate hospitalization hazards among the exposed without reflecting true morbidity. This could be especially true if, as shown in previous literature[13], CHIKF exposure is associated with preterm birth, which would prompt clinicians' closer observations. However, our results were consistent in the sensitivity analyses excluding preterm births. Also, no official guidelines recommend routine hospitalization or monitoring for CHIKF-exposed neonates in Brazil, making routine observation hospitalization less probable in the Brazilian context. Sixth, we could not obtain information on congenital infections by other pathogens, such as CMV and Toxoplasmosis. Newborns with other congenital infections may be included in both the CHIKF-exposed and non-exposed groups in our study. Seventh, our study population represents the poorer half of the Brazilian population; as a result, the findings may not be fully generalizable to the broader population or to higher-income settings. However, because the underlying mechanism is likely biological, we expect an increased risk to be observed in other contexts, warranting further investigation. Eighth, the Arboviruses reported in SINAN are often clinically diagnosed and are not serologically confirmed. We have used only the cases confirmed as CHIKF in subsequent epidemiological investigation[26], improving the precision of our exposure definition. However, our sensitivity analyses using laboratory-confirmed CHIKF cases lacked the statistical power to reproduce the results of our primary analyses, and misclassification bias cannot be denied entirely.

Despite these limitations, this study is timely and globally relevant, given the increasing frequency of CHIKF epidemics and the expansion of affected areas due to climate change. In the summer of 2025, CHIKF cases were reported in several EU countries, including France, Italy, and the UK, and these numbers are expected to rise in the coming years[2,3]. Our study highlights the importance of preventive measures, such as mosquito bite protection, particularly during pregnancy, as the effects may extend across generations. In Brazil, where epidemics are recurrent, these findings will help policymakers anticipate increased healthcare demands in the years following outbreaks.

In summary, we have shown that the hazard of under-three hospitalization was 21% higher among infants exposed to in-utero CHIKF, corresponding to 37 excess hospitalizations per 1000 exposed infants. Intrapartum infection was associated with a twofold higher hazard of hospitalization. Elevated hospitalization risks were also observed for infections occurring during the first and second trimesters. Evidence for the risk of death was limited.

## Methods

### Study design and setting
This is a longitudinal study using linked Brazilian administratively collected data for births between 2015–2018. Enrolled individuals were dynamically followed from birth until the first relevant event, up to age 3, or December 31, 2018, whichever occurred first. For hospitalization analyses, the follow-up period ended at the first hospitalization, with deaths considered as censoring events; for mortality analyses, the event was death.

The period of exposure to in-utero CHIKF extended from conception to delivery.

### Data source
The details of each dataset, including the variables obtained from the dataset and the linkage, are explained in Supplementary Material 1. Data linkage and cleaning were completed in 2023. Information on the maternal-child dyads was obtained from the CIDACS (Centro de Integração de Dados e Conhecimento para a Saúde) Birth Cohort, which links records from the Unified Registry for Social Programs (CADU), representing families who have applied for social benefits from the government[27], and the Live Birth Information System (SINASC)[28]. This cohort represents the poorest half of the Brazilian population[29].

The CIDACS Birth Cohort was linked to multiple national health information systems: the Information System for Notifiable Diseases (SINAN)[30], which contained the infection records of CHIKF, Dengue fever (DENF), and Zika fever (ZIKF); the Hospital Information System (SIH), which included registries of all hospitalizations financed by the Unified Health System; and the Mortality Information System (SIM)[31], which included registries of all deaths. This project is part of the Study of Early-Life Exposures During Developmental Stages (SEEDS) and has been approved by the Research Ethics Committee (CEP) of the Gonçalo Moniz Research Center (CAAE: 71054923.8.0000.0040) and LSHTM Ethics (Ref: 29560)[32].

### Study population
Eligible individuals were all singleton live-born infants born between January 1, 2015, and December 31, 2018, to women aged 15 to 49 years who were registered in CADU. Live-born was defined as a birth registered in SINASC with gestational weeks greater than 20 and a birth weight above 500 g. We excluded records with inconsistent dates (events occurring before the date of birth, or conflicting birth dates across registries); births with missing gestational age; births with weights above 6000 g; births with documented in utero exposure to ZIKV or DENV; and births with inconsistent CHIKF infection data or discarded in epidemiological investigation (details given in the exposure section).

### Exposures
CHIKF is a mandatory notifiable disease in Brazil, and all suspected and diagnosed cases are reported to the Epidemiological Surveillance Systems and recorded in SINAN[33]. In utero exposure to CHIKF in this study was defined as live births whose mothers were recorded as CHIKF cases recorded in SINAN and later confirmed by the epidemiological investigation. Live births not linked to any suspected CHIKF case in SINAN were classified as unexposed.

The exposure period was defined as the interval between the estimated date of conception and the date of delivery, with the timing of infection assessed using the self-reported date of symptom onset. The date of conception was estimated by subtracting the gestational age in weeks from the date of birth.

The Brazilian Ministry of Health guideline obligates an epidemiological investigation for all arboviral cases reported in SINAN by a designated team, to classify the reported case as either confirmed, inconclusive or discarded, based on clinical trajectory, local epidemiological trend or laboratory data when available[26]. We included only the cases confirmed as CHIKF, and the inconclusive and the discarded participants were excluded from the study population (detailed in Supplementary Material 2).

For individuals with more than one CHIKF notification in SINAN during the same pregnancy, the interval between the first and last registries was calculated. When multiple records were found within a 30-day period, they were considered part of the same infection episode, and the earliest record was used for analysis. If the interval between records exceeded 30 days, the individual was excluded from the analysis due to uncertainty about both episodes, as studies show that CHIKF infection usually confers lifelong immunity[34]. Individuals who had CHIKF records with a symptom onset before January 2015 were also excluded from the study population due to the uncertainty of the accuracy of the CHIKF diagnosis before its outbreak in 2015.

We conducted stratified analyses by the trimester in which the exposure occurred. The trimester of infection during pregnancy was

classified as the first (conception to 97 days), second (98 to 195 days), or third (196 days to birth)[35]. We further stratified exposure by whether the mother had an intrapartum infection to distinguish the potential for vertical transmission during maternal viremia close to delivery from the indirect effects of earlier infection. Intrapartum infection was defined as symptom onset within the seven days preceding delivery, based on evidence that CHIKF viremia typically lasts 7–10 days after symptom onset[36]. Studies from La Réunion have shown that, among parturient women, viral loads remain elevated for a relatively short period and that vertical transmission occurs only when symptom onset is within two days before or after birth[5]. We conducted a sensitivity analysis using this narrower definition, with details presented in the Sensitivity Analysis section.

### Outcome

Our primary outcome was all-cause first hospitalization under the age of three (2 years, 11 months, and 29 days). Although the hospitalization data for December 2018 were partially incomplete, we decided to retain the original censoring date of December 31, 2018, as a sensitivity analysis ending the follow-up period at the end of November 2018 yielded similar results. We selected a three-year follow-up period because, although linkage data were technically available up to age 4, only a small proportion of children, especially those in the exposed group, had reached the complete 4-year follow-up. We also described the proportions of hospitalizations by International Classification of Diseases (ICD-10) categories[37]. The secondary outcome was all-cause death under the age of three (until 2 years, 11 months, and 29 days). All-cause hospitalization and death in the neonatal period (<28 days after birth) were also analyzed as secondary outcomes. Because the outcomes were derived from national registry data, we assumed that loss to follow-up is negligible.

### Covariates

Based on the literature, we created a Directed Acyclic Graph (DAG) (Supplementary Fig. 2). We adjusted for the following variables. Maternal education[8]: no formal education, 1–3, 4–7, 8–11, and 12 or more years. Self-reported maternal race/ethnicity[4,24]: White, Black (Preto), Mixed (Pardo), Indigenous, and Asian. Maternal age[24]: 15–19, 20–34, and 35–49 years old. The adequacy of antenatal care visits, as a proxy for healthcare access[38] was classified according to Brazilian Ministry of Health guidelines: adequate for gestational age, one visit fewer than recommended, or two or more visits fewer than recommended. Number of previous gestations[8]: zero (no previous pregnancies), or one or more. Urbanicity: self-reported, urban or rural. Year of conception: derived from the calculation mentioned in the exposure section. In the Cox models, some variables were collapsed to improve statistical power: the education level of no formal education and 1–3 years was combined into a single category, and maternal race/ethnicity was combined into white or non-white. Biological sex assigned at birth was also recorded as male or female, and a sensitivity analysis was conducted stratified by sex. Gender information for the infants was not available.

### Statistical analysis

We conducted a matched cohort, with each exposed infant matched to 10 unexposed using nearest-neighbor matching by month of conception and the Immediate Geographic Region (Região Geográfica Imediata, RGI) of residence. The RGI is a group of 510 neighboring municipalities that share aspects such as healthcare access and education[39]. This matching method aimed to account for the precipitation-related CHIKF waves and regional heterogeneity. No a-priori sample size calculation was performed for this study; all eligible births from Brazil were included in the data. Because in-utero CHIKF exposure is rare and matching was done on fine strata (month of conception and RGI), we used up to 10 unexposed births per exposed to ensure sufficient comparators and maximize statistical precision.

For the basic demographics, summary statistics were presented for the overall study population and by exposure status. Missing data were minimal and considered unlikely to be associated with the outcomes of interest (Supplementary Table 11); therefore, a complete-case analysis was conducted. 13.9% of eligible participants were excluded before matching due to missing covariate values. The difference in the distribution of covariates was measured with standard mean differences (SMD).

For the survival analyses, we used the Kaplan-Meier method and a multivariable Cox proportional hazards model, adjusted for confounders and stratified by matched groups. The Cox model used months as the time scale and estimated hazard ratios (HRs) for hospitalization and death by CHIKF exposure status, representing the relative incidence rate per person-month. The proportional hazards (PH) assumption was checked by visual inspection and using the Schoenfeld Residuals. We also estimated the marginal standardized absolute risk difference (RD) using the same stratified Cox model for outcomes at ages 12, 24, and 36 months. The RD represents cumulative risks per 1000 infants at chosen months. To account for the dependence introduced by matching, bootstrapping (B = 500) was performed within matched groups, with replacement, to calculate percentile-based 95% confidence intervals (CIs) for the HR and RD.

To investigate whether the magnitude of association differs by trimester of infection, we ran a Cox model with a four-level categorical exposure variable indicating the timing of in utero CHIKF exposure: unexposed (reference), first, second, or third trimester. The hazards for infection in the intrapartum period were similarly investigated.

We also conducted a sub-analysis examining hospitalizations and deaths that occurred during the neonatal period as the outcome. For this Cox model, individuals who did not experience the event were censored at 28 days after birth or December 31st, 2018, whichever came first.

### Sensitivity analyses

We conducted four sensitivity analyses, with the third and fourth included post hoc following the revision. The first analysis included only individuals born at term (37 weeks of gestation or later) and with normal birth weight (>2500 g). This analysis investigated whether the association between in-utero CHIKF exposure and hospitalization could be partially explained by prematurity or low birth weight. Given that a previous study shows an elevated risk of prematurity among CHIKF-exposed newborns[13], early hospitalizations among exposed infants could reflect precautionary observation rather than morbidity; this analysis was therefore intended to assess the potential contribution of such indication bias. The second sensitivity analysis included only the lab-confirmed CHIKF cases and their matched controls. Previous studies report that DENF and CHIKF may be misclassified because of their similar clinical presentations; thus, we aimed to identify a more accurate population of exposed individuals[40]. The third sensitivity analysis applied an alternative definition of intrapartum exposure based on evidence from La Réunion studies[5], classifying infants as having intrapartum exposure if maternal symptom onset occurred between two days before and two days after delivery.

The fourth sensitivity analysis was stratified by sex. For this analysis, we included the exposed in one sex and their matched pair with the same sex. Because the death event was sparse for both sexes (12 in males and 26 in females), we only performed the sex-stratified analyses for hospitalization.

All analyses were performed using R (versions 4.2.1 and 4.4.1), with the MatchIt package for matching and the survival and survminer packages for survival analyses. We have reported our findings in accordance with the STROBE guidelines (Supplementary Table 12).

**Reporting summary**

Further information on research design is available in the Nature Portfolio Reporting Summary linked to this article.

## Data availability

The relevant data are included in the manuscript and appendix. Deidentified individual participant data are available on reasonable request to CIDACS, subject to institutional collaboration agreements and Brazilian ethics committee approval.

## Code availability

No custom code was developed; analysis was conducted using publicly available R packages. Code is available upon request.

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

## Acknowledgments

EP receives funding from Wellcome Trust 225925/Z/22/Z. TC-S receives funding from the Royal Society (NIF\R1\231435).

## Author contributions

M.K. and E.P. conceived and designed the study and developed the protocol. M.K., O.C., and T.C.-S. performed the data analysis. M.K. drafted the manuscript. M.K., O.C., T.C.-S., V.B., M.G.T., M.B., and E.P. contributed to the interpretation of data and critically reviewed and approved the final version of the manuscript. E.P. supervised the study. M.K., O.C., T.C.-S., and E.P. had full access to all the data and took responsibility for the integrity of the data and the accuracy of the data analysis.

## Competing interests

The authors declare no competing interests.
