## [Transparent Peer Review file · Nature Communications]

In-Utero Exposure to Chikungunya and Child Morbimortality: A Population-Based Study Using Linked Routine Data

Corresponding Author: Dr Mio Kushibuchi

Version 0:

Reviewer comments:

Reviewer #1

(Remarks to the Author)

Manuscript Number: NCOMMS-25-77120

Type: Research article

Title: In-Utero Exposure to Chikungunya and Child Morbi-mortality: A Population-Based Longitudinal Study

General Comment

Documenting the long-term outcomes of chikungunya infection in areas endemic to virus (CHIKV), or where it is epidemic, is key to enhance epidemiological surveillance and improve its prevention, given its potential for lifelong consequences and related disabilities, both in the neonate and the developing child, especially as result of mother-to-child vertical transmission of the virus. In this framework, after publishing a first study comparing the perinatal outcomes of symptomatic chikungunya, dengue and zika infections in the journal (Cerqueira-Silva T et al, Nat Comm 2025), the authors report on the three-year child outcomes of chikungunya infection. Interestingly, they observe a 22% increase in the hazard of hospitalization, overall, and when the onset of maternal infection occurred within the first or second trimester of gestation, or when the mother has been infected intrapartum (HR 1.98); they also observe a higher risk of neonatal death, when the mother has been infected intrapartum (HR 3.91). These are novel and important findings that complement previous reports from the same group in the 100 Million Brazilian cohort, 2015-18, on increased all-cause mortality within 3 months post-infection, or increased risk of deaths secondary to diseases onset for cerebrovascular diseases, diabetes, or ischemic heart diseases within 28 days (Cerqueira-Silva T et al, Lancet Infect Dis 2024), or from Dr Freitas, a militant epidemiologist, champion on excess mortality data in chikungunya epidemics (Freitas ARR et al, PLoS Current 2017; Freitas ARR et al, Trans R Soc Trop Med Hyg 2018; Freitas ARR et al, Epidemiol Infect 2018; Freitas ARR et al, Emerg Infect Dis 2018; Freitas ARR et al, Pathog Glob Health 2019; Frutuoso LCV et al, Rev Doc Bras Med Trop 2020). Altogether, these data have the potential to change the regard on chikungunya virus, a virus formerly known to cause mild, self-limiting illness, and shed light on its potential to cause harms and deaths, which represent a game changer and deserves publication. The paper is well-written, very rich in results and goes deep in interpretations, its methods without major flaws. However, the data linkage of the different sources that ensure statistical power may be also a source of limitations because it does not allow the acquisition of detailed ad hoc data to support finer interpretations. For instance, it is well known in perinatal medicine that known risks of adverse outcomes lead to hospitalization for surveillance without the need to observe the disease but rather to prevent it or limit its consequences. This usually leads to overestimation of the outcome event. In the same manner, the fact of ignoring the exact risk of maternal-fetal or maternal-neonatal transmission situations does not allow attribute neonatal excess mortality to the infection as it may also result from anticipatory iatrogenesis of inappropriate obstetrical practices. It is therefore necessary to temper the interpretation of certain results so as not to mislead readers, because some results may proceed from indication biases that are undetectable by this type of study, or they may reflect suboptimal perinatal indicators (in Brazil, maternal mortality and infant mortality rates are six times and three times higher than in France, respectively), which would make the results non generalizable. It should be also interesting to report on a fifth sensitivity analysis excluding preterm births and neonatal death from intrapartum transmission to control indication bias and better assess the risk of hospitalization.

Specific comments

Title and abstract

1(a). Indicate the study's design with a commonly used term in the title or the abstract. I would add the notion of data linkage, not to misguide the reader, as this is not a cohort. For example: "In-Utero Exposure to Chikungunya and Child Morbi-

mortality: A Population-Based Longitudinal Study through data linkage.”

1(b). Provide in the abstract an informative and balanced summary of what was done and what was found.

Page 2, line 20. “The long-term effect of in utero-exposure is not known.” This statement may apply for Brazil and similar countries, but French authors have reported longer term neurodevelopmental outcomes in the follow-up of the CHIMERE cohort study (Sarton R et al, *Viruses* 2025), while, in France, a higher standard-care country, the outcome of in utero-exposed uninfected children was the same that that of unexposed-uninfected children (Fritel X et al, *Emerg Infect Dis* 2010; Gérardin P et al, *PLoS Negl Trop Dis* 2014). Please, revise to propose a more contrasted sentence.

Page 2, line 31. With respect to infant mortality, the interpretation overstates the findings, as it only stands for intrapartum exposures, and we have no background information on maternal-neonatal vertical transmission nor on iatrogenic or suboptimal perinatal care, to link the infant death to the infection directly or indirectly.

Introduction

2. Background/rationale

Explain the scientific background and rationale for the investigation being reported.

Background science is somewhat biased toward confirmation and preservation of Brazilian findings.

Page 3, line 43. “Current evidence on the effects of in utero CHIKF exposure on newborns remain inconsistent.” Neonates have long been designated as an at-risk population for severe chikungunya. Please, add some antagonist references to support the statement.

Page 3, lines 43 to 45. “Most studies do not report associations with adverse birth outcomes; however, in cases of vertical transmission, severe neonatal transmission has been documented.” We also expect some references here. Please quote some of the seminal studies that have first report on these background data. There are also more recent studies in Vietnam or Nigeria that have report on birth outcomes.

Page 3, lines 45 to 47. “A systematic review of 266 in utero-exposed report that vertical transmission occurs in up-to 48.7% of intrapartum exposed cases...” That is true however the vertical transmission rate is very difficult to assess given the unreliability of umbilical cord blood antibodies, placenta samples, and lack of studies controlling postnatal PCRs in neonates. The risk vertical transmission rate and fetal mortality risk have also been meta-analyzed by north American authors, this should be quoted here along with the seminal study driving these background data to avoid the abovementioned cognitive biases.

Page 3, lines 49 to 51. “However, more recently, a nationwide study from Brazil involving 6,000 exposed newborns found that second-trimester CHIKF infection was associated with increased risks of low Apgar scores and neonatal death.10” That is true that large powered studies are needed to report on scarce outcomes but the statement is incomplete. The study also reported an association between symptomatic maternal chikungunya and increased risk of preterm birth (Cerqueira-Silva T et al, *Nat Comm* 2025). The authors should elaborate here on causality discussing perinatal care in the light of Brazilian perinatal indicators and be more sincere and quote explicitly that is a study by their group. There should be no notoriety bias in reporting novel findings on valuable data.

Pages 3 to 4, lines 52 to 54. “Evidence on the long-term consequences of in utero-exposure is scarce. A study from La Réunion has prospectively followed 33 children for two years, revealing higher prevalence of global neurodevelopmental delay (GND) in the exposed...” That is poor review of the literature and inadequate reporting. For sure, since big countries are involved in chikungunya research, it is very difficult to report updates of Reunionese cohorts with longer term outcomes (Sarton R et al, *Viruses* 2025; Sarton et al, *SSR preprint* 2025). For sure, the CHIMERE study may seem small in comparison to their study, but this was indeed a cohort study set up during a real time epidemic, and this study did not report on GND prevalence, but on GND cumulative incidence, as it used incidence rate ratios. The authors should provide the readership with the most recent evidence and quote the literature appropriately.

3. Objectives

State specific objectives, including any prespecified hypotheses.

Page 4, lines 57 to 60. Hypotheses are missing.

Methods

4. Study design

Present key elements of study design early in the paper.

Page 4, lines 63 to 65. The study is indeed longitudinal through data linkage of several valuable, well-powered databases, but this is not a real prospective cohort study providing ad hoc fine data susceptible to argue for causality beyond a literature perspective. The data were gathered from several prospective routinely collected surveillance data.

5. Setting

Describe the setting, locations, and relevant dates, including periods of recruitment, exposure, follow-up, and data collection

6. Participants

6 (a). Cohort study—Give the eligibility criteria, and the sources and methods of selection of participants. Describe methods of follow-up.

Page 5, lines 87 to 88. It is written “birth with gestational age less than 20 weeks; birth with weights below 500 gr...”. These are no births, as in perinatal medicine the term birth is reserved for deliveries at gestational age or birthweights compatible with viability. Before 20 weeks or below 500 gr, the authors should prefer late miscarriage or early fetal losses.

6 (b). Cohort study—For matched studies, give matching criteria and number of exposed and unexposed.

The authors provide sufficient information about their matching methods but supplementary material 4 should give p values of comparisons between matched and non-patched participants to let the reader know if there could result selection bias from matching.

7. Variables

Clearly define all outcomes, exposures, predictors, potential confounders, and effect modifiers. Give diagnostic criteria, if applicable

Page 6, line 94. The authors should report more information on SINAN exhaustiveness and representativeness, to allow more comprehension.

Page 6, lines 107 to 109. The definition of the intrapartum exposure to the infection is spurious. It is based on a very partial

narrative review from Brazilian authors, which endorses a lasting CHIKV viremia of 7-10 days, as found in textbooks. This applies in most nonpregnant adults but not for pregnant women. Indeed, it was shown in a collaboration between Reunion island perinatal specialists and Institut Pasteur researchers, that acute symptoms and the infectious viral loads in 61 parturient women were short-lived (up to 5 days) and needed to peak between day-2 and day+2 around delivery to infect the placenta and being able to be transmitted vertically to the fetus (Gérardin P et al, PLoS Med 2008). In addition, regular and intense uterine contractions needed to be observed to define the intra-partum period, (pre-labour and labour). In accordance, we recommend to discuss the possibility of misclassification using an epidemiological definition and the opportunity to reclassify women (who were indeed in the prepartum period) from the intrapartum to the antepartum group.

8. Data sources/ measurement

For each variable of interest, give sources of data and details of methods of assessment (measurement). Describe comparability of assessment methods if there is more than one group

9. Bias

Describe any efforts to address potential sources of bias, especially indication bias for hospitalization, as in most obstetrical guidelines pregnant women with fever should be referred to the hospital or even, treated as inpatient. In the same manner, discuss indication bias for hospitalization the intrapartum period, as a 50% (or 48.7%) absolute risk of vertical transmission rate followed by a 50% risk of severe neonatal infection is a powerful incentive to monitor neonates with alarms in neonatal care units, at least for three days, until blood virologic samples are negative. Indeed, a negative PCR at day+3 is often reassuring in daily pediatric clinical practice and sometimes used to allow the mother back home with her newborn child, without a risk of developing symptoms.

10. Study size

Explain how the study size was arrived at, especially why matching 1 infected neonate with 10 unexposed neonates, and more or less ?

11. Quantitative variables

Explain how quantitative variables were handled in the analyses. If applicable, describe which groupings were chosen and why

12. Statistical methods

12 (a). Describe all statistical methods, including those used to control confounding.

Page 8, lines 140 to 141. Justify the preference of standard mean differences (SMD) to compare the distribution of covariates between exposed and nonexposed groups. SMD are less intuitive than p values for non-epidemiologist readership and require normality to gain over p values.

12 (d). Cohort study—If applicable, explain how loss to follow-up was addressed

12 (e). Describe any sensitivity analyses.

Page 9, lines 164 to 165. The “intubation period” is not clear for readers non familiar with perinatal dengue infection or perinatal medicine. The authors should refer to the abovementioned paper (Gérardin P et al, PLoS Med 2008) to justify the extension of the intrapartum period to day+2, as in real life the acme of maternal symptoms may be postponed in postpartum. Add a fifth sensitivity analysis excluding preterm births and neonatal death from intrapartum transmission to control indication bias and better assess the risk of hospitalization.

Results

13. Participants

14. Descriptive data

14 (b). Indicate number of participants with missing data for each variable of interest

14 (c). Cohort study—Summarise follow-up time (eg, average and total amount)

Table 1. All SMD values are small (< 0.5) except that of year of conception that is within the medium range (0.5 to 0.8). The authors should provide additional information on associated p values, for example * p< 0.01-0.05 ** p< 0.001-0.01 *** p<0.001 to help interpret SMD values.

Page 10, lines 182 to 184. “The exposed group had a higher percentage of individuals residing received adequate antenatal care (83% versus 78%, SMD 0.136)...” These findings are counter intuitive. We expect more impoverished people in the exposed group, who are gathered in urban favellas, but it surprising that these people have more antenatal care. This could reflect both an indication bias and an information bias, those having more antenatal care visits including additional visits due to exposure to the CHIKV. This could also reflect Brazilian health promotion programs before 2019 directed towards reduction of social inequalities in health. Please, explain.

15. Outcome data

15 (a). Cohort study—Report numbers of outcome events or summary measures over time.

Page 11, lines 187 to 189. 687 or 686 (Suppl. Mat 5) ? 3157 or 3154 (Table 1, Suppl Mat 5) ? 5705 or 5619 (Suppl Mat 5) ? 380 or 379 (Suppl Mat 5).

Supplement material 5: Hospitalizations. The sum of neonates exposed in first, second and third trimester gestation (281+214+215) is over the total number of exposed (686 or 687). Check, revise or explain.

16. Main results

16 (b). Report category boundaries when continuous variables were categorized

17 Other analyses

Report other analyses done—eg analyses of subgroups and interactions, and sensitivity analyses.

Page 12, line 219. The term “consistent” is a little strong. Check and temper, please.

Page 13, line 224. The data are not shown. Please, check and provide the data in supplement material or add “(data not shown)”.

Discussion

18. Key results

Summarise key results with reference to study objectives

Page 13, line 233. “Our study is the first to measure health outcomes with a three-year follow-up period.” Contains an

unnecessary a priority claim (i.e., This the first study to assess, or we are the first to describe). Instead of generating interest in a study, this claim tends to trivialize the findings. In addition, such claims may offend authors whose earlier papers on the topic may have appeared elsewhere. We strongly suggest that you revise the phrase to include the word “novel” or “unique” even though Reunionese studies have similar long and even longer hindsight.

Page 13, line 238. “A study from Brazil...” is to be replaced by “A previous study by our group.” Which is more appropriate.

19. Limitations

Discuss limitations of the study, taking into account sources of potential bias or imprecision. Discuss both direction and magnitude of any potential bias, especially indication bias for hospitalizations in the intra partum period (see above). Also, discuss this bias in the light of the fifth requested sensitivity analysis.

20. Interpretation

Give a cautious overall interpretation of results considering objectives, limitations, multiplicity of analyses, results from similar studies, and other relevant evidence

Page 13, line 241. The authors assimilate hospitalizations automatically as complications but admissions to the hospital may also result from an indication bias (need for close monitoring of healthy neonates at-risk as born with a mother positive for CHIKV in the intrapartum period).

21. Generalisability

Discuss the generalisability (external validity) of the study results

Page 14, line 257. “Evidencing that CHIKF can cause serious harms to the fetus highlights the need for preventive measures...” Causal inference cannot be claimed from this study, only presumptions.

Additional comments

Ref#10 lacks year of publication; issue (volume) and pagination.

Reviewer #2

(Remarks to the Author)

This is a well-designed and policy-relevant study that addresses an important evidence gap regarding the long-term consequences of in-utero Chikungunya virus (CHIKV) exposure on child morbidity and mortality. The use of a large, population-based cohort, robust data linkage, and matched design makes this work both timely and valuable for public health audiences in endemic and emerging CHIKV regions. The manuscript is generally well written and methodologically sound. My comments below focus on clarifying methodological details, improving transparency, and strengthening the interpretation of results and discussion of limitations.

Introduction

The Introduction is logically structured and timely, highlighting the public health relevance of Chikungunya virus (CHIKV) and the lack of longitudinal evidence on maternal infection during pregnancy. It can be improved by the following minor revision:

- Line 44: Early clarification of the definition of in-utero CHIKF exposure (i.e., maternal CHIKV infection during pregnancy), this is vital to align the term with the Methods and avoid confusion.
- Line 44: Typo: “remains inconsistent.”
- Line 59: The Introduction should explain briefly why hospitalization and death up to age three were chosen to represent “long-term morbidity and mortality.”

Methods

The Method is generally comprehensive and clearly described, using robust national datasets and appropriate linkage procedures. The matched cohort design is well justified. However, several points need clarification or refinement to strengthen transparency and reproducibility:

• Study population:

Lines 86–90 – Depending on the electronic health records, have you considered the continuity of registration before conception (for data completeness) or deregistration of infants or young children due to migration (for censoring)?

• Exclusion criteria and exposure classification:

Lines 101–103 – The exclusion of women with >30-day gaps between CHIKF notifications should be supported by references confirming that reinfection within this interval is biologically implausible.

Lines 106–108 – Please also clarify the reason for the trimester classification (conception–97 days, 98–195 days, 196 to birth).

Lines 109–111 – Intrapartum exposure: Add a brief explanation for stratifying intrapartum infections, e.g., “to distinguish probable vertical transmission during maternal viremia near delivery from indirect effects of earlier maternal infection.”

• Study design and follow-up:

The study design clearly describes the follow-up window (until age 3, 31 December 2018, or the outcome event, whichever occurred first). However, the primary outcome was first hospitalization up to age 2 years 11 months and 29 days; rephrasing this for consistency would improve clarity.

The rationale for choosing a three-year follow-up should also be briefly explained (e.g., data completeness or early-childhood morbidity relevance).

• Covariate adjustment and DAG:

The DAG is informative, but as maternal socioeconomic status (SES) appears to cause urbanicity, adjusting for both may be redundant. Once SES is included, the backdoor path through urbanicity is already blocked; further adjustment may reduce precision without addressing additional confounding.

- Missing data:

Lines 139–141 – Complete-case analysis is acceptable as long as the limitation is acknowledged in the Discussion. However, the overall exclusion proportion (~13.9% of eligible participants) should also be reported. Distinguishing between variable-level and individual-level missingness would improve transparency.

- Time scale and risk measures:

Lines 144–150 – Clarify that months were used as the time scale, meaning hazard ratios represent relative incidence rates per person-month, and that standardized risk differences represent cumulative risks per 1,000 children at 12, 24, and 36 months.

- Sensitivity analyses:

Sensitivity analyses are currently spread across several sections. Consolidating them into a single “Sensitivity analyses” subsection would improve structure and readability.

Results

- Table 1 (Outcome labels):

The labels “All-cause hospitalization until age 3” and “All-cause death until age 3” should be revised to reflect that these represent the first hospitalization/death during follow-up, with censoring at the third birthday or 31 December 2018.

- Table 1 (SMD):

Define “SMD” in the table footnote as “Standardized Mean Difference” for clarity.

- Figure 2 (Kaplan–Meier curves):

The x-axis should be labelled “time since birth (months)” to clarify that it represents follow-up time to first hospitalization or death.

- Figures 3 and 4 (Sample size):

The figure titles indicate N = 39,694, but the analytic cohort throughout the manuscript is N = 34,694. This should be corrected for consistency.

- Figure 3 (Subgroup sample sizes):

The figure would be more informative if the number of participants per subgroup (currently reported in Supplementary Material 5) were incorporated directly into the figure. Presenting these counts alongside each HR would improve interpretability and show the precision of subgroup estimates.

- Results text (Interpretation of findings):

Replace terms such as “statistically significant” or “not significant” with phrasing that reflects the strength of evidence (e.g., “strong/moderate/little evidence of an association”), consistent with Nature’s reporting standards and best epidemiological practice.

- Results text (Absolute risk difference):

Clarify that the absolute risk difference (32.9 per 1,000) represents the cumulative risk difference up to 36 months of follow-up (not the incremental difference between 24–36 months). Wording such as “over the first three years of life” would improve interpretability.

- Terminology (throughout manuscript):

The study population is repeatedly described as “children,” but given that follow-up extends only to age three, “infants” (or “infants and young children”) would be more accurate.

Discussion

The Discussion is generally well structured, with logical interpretation of findings and thoughtful consideration of biological mechanisms. Several aspects could be clarified or expanded to strengthen interpretability, transparency, and contextual relevance.

Generalisability

- Population representativeness:

The Discussion should explicitly acknowledge that the CIDACS Birth Cohort represents the poorest half of the Brazilian population and thus findings may not generalise to higher-income or nationally representative populations.

- Matching design:

The matching strategy by month of conception and Immediate Geographic Region (RGI) improves internal validity but limits external generalisability to births within similar time–place strata. This should be noted when discussing policy or public health implications.

Potential sources of bias

- Complete-case analysis:

The Discussion should acknowledge that approximately 13.9% of eligible individuals were excluded due to missing covariate data. If missingness was related to SES, healthcare access, or region, selection bias is possible. Clarifying this would strengthen the transparency of limitations.

Reviewer #3

(Remarks to the Author)

This manuscript addresses a highly relevant and timely public health issue: the effects of in utero exposure to Chikungunya virus (CHIKV) on child morbidity and mortality. The authors present a well-designed, population-based matched cohort study using a robust national dataset from Brazil. The manuscript is clearly written, adheres to STROBE guidelines, and provides transparent methodology and supplementary data.

The study’s strengths include its large sample size, rigorous matching procedures, and comprehensive sensitivity analyses. The findings—particularly the increased risk of hospitalization following intrapartum exposure—are compelling and of global relevance given the expanding geographic range of CHIKV due to climate change.

However, several aspects of the manuscript could benefit from further clarification, additional discussion, and minor

revisions to improve interpretability and scientific rigor.

Major Comments:

1) Confounding and Effect-modifiers

The introduction would benefit from a more explicit discussion of the limitations in previous studies, particularly regarding confounding and effect-modifiers. For example, how have prior studies failed to account for socioeconomic status, access to healthcare, or co-infections?

The authors should elaborate on how their matched cohort design mitigates these biases and clarify any residual confounding that may remain (in the methods or discussion).

2) Exclusion Criteria and Bias

The rationale for excluding births with ZIKV and DENV exposure is clear, but the manuscript does not mention other congenital infections (e.g., CMV, toxoplasmosis) or congenital anomalies. These are known risk factors for hospitalization and neonatal mortality. Were these considered or excluded? If not, how might they have influenced the results?

3) Live Birth Bias

The authors acknowledge the limitation of excluding stillbirths and miscarriages. This live birth bias is particularly relevant for infections occurring in the first trimester. A more detailed discussion of how this may underestimate the true burden of CHIKV exposure in early pregnancy would strengthen the manuscript (miscarriage, birth defects, early fetal demises, early placental abruption, ...).

4) Intrapartum Exposure

The finding that intrapartum exposure is associated with an increase in hospitalization risk is striking. The authors should expand on the biological plausibility of this result. For instance, could the severity be due to direct vertical transmission during delivery, immature neonatal immune response, lack of maternal IgG transfer, or higher viral loads?

Minor Comments and Suggestions:

Line 46: Consider rephrasing to "severe neonatal morbidity and long-term sequelae have been documented in cases of vertical transmission."

Line 48: Replace "sepsis" with "CHIKF clinical syndrome" to avoid confusion with bacterial sepsis.

Line 85: Clarify whether congenital infections and birth defects were excluded or adjusted for.

Line 267: Expand on the limitation regarding underreporting of early pregnancy events, especially in low-resource settings.

Discussion: Add a brief explanation of why intrapartum infections may be more severe (e.g., timing of exposure relative to delivery, lack of maternal antibody transfer).

Version 1:

Reviewer comments:

Reviewer #1

(Remarks to the Author)

Manuscript Number: NCOMMS-25-77120R1

Type: Research article

Title: In-Utero Exposure to Chikungunya and Child Morbi-mortality: A Population-Based Longitudinal Study

General Comment

I thank the editor for allowing me to re-assess this important manuscript.

The authors have done their best for addressing most of my remarks.

I have no additional claim that could be imperative at this step and I think the paper could be published after give some precisions asked in the Specific comment section.

Specific comments

With respect to infant mortality. I agree that the standard of neonatal care is not different between the exposed and the nonexposed children, yet perinatal chikungunya adds an excess mortality risk, due to escalating obstetric care for acute fetal distress. The risk does not exist in other situations of perinatal asphyxia where obstetric care is better codified and better managed by experience. Reduced iatrogenic effects during the perinatal period in the nonexposed group may affect the risk of subsequent hospitalization beyond the neonatal period differentially. Inversely, a selective survival bias may affect the exposed group that survived the neonatal period likely affecting hospitalization rates. Indeed, the balance between the two trends is highly unpredictable with still the possibility to affect the estimates without not necessarily skewing them. This requires great caution in interpretations, but no revision at this step of the publication.

Introduction

2. Background/rationale

"Evidence on the long-term consequences of in utero exposure is scarce"

Page 4, lines 9 to 11. I agree with the statement, however the outcomes of long-term consequences are very different according to what is meant by exposure. The exposure is qualitatively very different in the 4 studies cited. The Grenadian

study (ref#14) reports on neonates that were exposed antenatally, however none was proven infected (the infected term was used for children with positive CHIKV-specific IgG antibodies which indeed indicates maternal infection with passive transfer of transplacental antibodies; or children with IgG+/IgM+ status that indicated postnatal infection). The Brazilian study (ref#15) reports quasi exclusively on mothers infected antenatally or before conception with only one case exposed perinatally who suffered acute fetal distress but got not infected. This study likely suffers a “Zika background bias.” The Curaçao paper (ref#16) refers to both perinatal (n=5) and postnatal (n=17) infections (this is specified in both the abstract and the text) but the seven cases of cognitive delays are observed in postnatally infected children (cf discussion). The Indian paper (ref #17) refers to 13 children born “extramurally” without providing information on mothers nor on confirmatory diagnosis (no child pcr or CHIKV-specific serology) or the lag between birth and presentation to the hospital. The information can be found in a companion paper by the same team (Maria A, et al, Indian Pediatr 2018) which precises that the infection was acquired through perinatal transmission (n=10), postnatal transmission (n=2) or the timing was unknown (n=1). I would suggest to remove the Curaçao paper and rather to consider “the findings are mixed” to write that evidence is increasingly showing an increased risk of adverse neurodevelopmental outcomes adding the earlier contribution of the CHIMERE cohort study (Gérardin P et al, PLoS Negl Trop Dis 2014) and the last Brazilian contribution by the Rio cohort (Pinho de Almeida Di Maio Ferreira FC et al, J Pediatr 2015). Both provide similar conclusions on early neurodevelopmental outcomes with very similar sample sizes.

Page 4, line 11. I would add “Novel” to “Evidence from La Réunion”... Unfortunately our last paper which reports now on 13-15 year outcomes is still in the limbs (Sarton R et al, SSRN preprint 2024),

3. Objectives

State specific objectives, including any prespecified hypotheses.

Page 4, line 17. I would specify in the hypothesis “beyond the neonatal period” as exposed children may be hospitalized in their first days of life just for surveillance and monitoring the onset of symptoms.

Methods

6 (b). Cohort study—For matched studies, give matching criteria and number of exposed and unexposed.

Matching methods.

I agree with the idea that p-values are highly dependent on the sample size. I understand that the authors having a large sample size have preferred reported standardized mean differences (SMDs), as routinely done in meta-analyses, large RCTs and large observational studies.

It is true that a SMD < 0.1 is often taken for defining perfect balance, however it has also been shown controversial.

Conversely, for several authors there is no strict consensus on which SMD value (>0.1, >0.20 or >0.25) define that matching is unbalanced (Ho DEK et al, Political Analysis 2007; Harder VSE, Psychol Methods 2007; Hedges LV, Educ Psychol Meas 2024; Harden JJ 2025). It is also known that SMD can falsely declare studies as biased when sample size is too small or too large (Austin PC et al, Stat Med 2009). Similarly, there is no consensual method to balance diagnostics after propensity score matching, that means correcting the imbalance between the treated and untreated group (Zhang Z et al, Ann Transl Med 2019).

As the authors provided p-values in their rebuttal latter, I would suggest reporting SMDs in both Table 1 and supplementary material 4 and adding a p-values column in supplemental material 4.

In this document, the columns could be as follows:

Not matched Overall Matched Matched p-values SMDs
Nonexposed Nonexposed exposed

This will suggest that matching is well balanced as SMDs are never large (> 0.8) and that except for three variables (number of ANC, urbanicity of residence, and year of conception), very significant p-values are often consistent with small SMDs, as a consequence of the large sample size.

9. Bias

I acknowledge the authors have provided substantial efforts but I still don't understand why the authors do not wish to provide the demanded sensitivity analyses, especially in preterm birth, or why they might introduce a survival bias, while results seem consistent for hospitalization. Do they mean a selective survival bias ? Maybe these elements should be introduced in the Method sections and/or discussion, because this will likely question physicians unfamiliar with Brazilian standard cares in obstetrics and perinatal medicine.

Reviewer #2

(Remarks to the Author)

I thank the authors for their careful and comprehensive responses to my comments. The revisions have substantially improved the clarity, transparency, and interpretation of the manuscript. I have no further concerns and support publication.

Reviewer #3

(Remarks to the Author)

The authors responded accurately and appropriately to all of my comments and amended the manuscript accordingly. I have no further comments or suggestions.

REVIEWER COMMENTS

Reviewer #1 (Remarks to the Author):

Manuscript Number: NCOMMS-25-77120

Type: Research article

Title: In-Utero Exposure to Chikungunya and Child Morbidity-mortality: A Population-Based Longitudinal Study

General Comment

Documenting the long-term outcomes of chikungunya infection in areas endemic to virus (CHIKV), or where it is epidemic, is key to enhance epidemiological surveillance and improve its prevention, given its potential for lifelong consequences and related disabilities, both in the neonate and the developing child, especially as result of mother-to-child vertical transmission of the virus. In this framework, after publishing a first study comparing the perinatal outcomes of symptomatic chikungunya, dengue and zika infections in the journal (Cerqueira-Silva T et al, Nat Comm 2025), the authors report on the three-year child outcomes of chikungunya infection. Interestingly, they observe a 22% increase in the hazard of hospitalization, overall, and when the onset of maternal infection occurred within the first or second trimester of gestation, or when the mother has been infected intrapartum (HR 1.98); they also observe a higher risk of neonatal death, when the mother has been infected intrapartum (HR 3.91). These are novel and important findings that complement previous reports from the same group in the 100 Million Brazilian cohort, 2015-18, on increased all-cause mortality within 3 months post-infection, or increased risk of deaths secondary to diseases onset for cerebrovascular diseases, diabetes, or ischemic heart diseases within 28 days (Cerqueira-Silva T et al, Lancet Infect Dis 2024), or from Dr Freitas, a militant epidemiologist, champion on excess mortality data in chikungunya epidemics (Freitas ARR et al, PLoS Current 2017; Freitas ARR et al, Trans R Soc Trop Med Hyg 2018; Freitas ARR et al, Epidemiol Infect 2018; Freitas ARR et al, Emerg Infect Dis 2018; Freitas ARR et al, Pathog Glob Health 2019; Frutuoso LCV et al, Rev Doc Bras Med Trop 2020). Altogether, these data have the potential to change the regard on chikungunya virus, a virus formerly known to cause mild, self-limiting illness, and shed light on its potential to cause harms and deaths, which represent a game changer and deserves publication. The paper is well-written, very rich in results and goes d

deep in interpretations, its methods without major flaws. However, the data linkage of the different sources that ensure statistical power may be also a source of limitations because it does not allow the acquisition of detailed ad hoc data to support finer interpretations. For instance, it is well known in perinatal medicine that known risks of adverse outcomes lead to hospitalization for surveillance without the need to observe the disease but rather to prevent it or limit its consequences. This usually leads to overestimation of the outcome event. In the same manner, the fact of ignoring the exact risk of maternal-fetal or maternal-neonatal transmission situations does not allow attribute neonatal excess mortality to the infection as it may also result from anticipatory iatrogenesis of inappropriate obstetrical practices. It is therefore necessary to temper the interpretation of certain results so as not to mislead readers, because some results may proceed from indication biases that are undetectable by this type of study, or they may reflect suboptimal perinatal indicators (in Brazil, maternal mortality and infant mortality rates are six times and three times higher than in France, respectively), which would make the results non generalizable. It should be also interesting to report on a fifth sensitivity analysis excluding preterm births and neonatal death from intrapartum transmission to control indication bias and better assess the risk of hospitalization.

We are very grateful for the reviewer's thoughtful and generous overall assessment of our work, and for situating our study within the broader body of research on chikungunya, including the important contributions from our own previous studies, from Dr Freitas and colleagues, and from the La Réunion research teams. We fully share the reviewer's view that documenting the long-term consequences of CHIKV infection—particularly in the context of mother-to-child transmission—is crucial for surveillance, prevention, and clinical care, and we appreciate the recognition that our findings add to this evolving picture.

In response to the reviewer's overall concerns, we have implemented several substantive changes throughout the manuscript. First, we have clarified the study design, explicitly describing it as a population-based longitudinal study based on linkage of routinely collected registries, and we have removed language that could imply a purpose-built prospective cohort or causal inference. Second, we broadened and rebalanced the Introduction and Discussion to more fully acknowledge prior work, especially the La Réunion and other international cohorts, and to clearly describe how our findings provide unique contributions using data linkage. Third, we have tempered the interpretation of hospitalization and mortality outcomes by explicitly discussing

indication bias and the limitations of our data, specifically, the lack of virological confirmation and detailed perinatal clinical information, and by adding sensitivity analyses based on the virological intrapartum window proposed by the La Réunion team. We also explored the possibility of a sensitivity analysis excluding preterm births and early neonatal events because of the high risk of survival bias; however, these limitations and their implications for generalisability are now clearly discussed. Finally, we revised wording throughout the abstract, results, interpretation, and generalisability sections to avoid causal language and to emphasize associations, contextualizing our findings against Brazilian perinatal indicators.

Specific comments

Title and abstract

1(a). Indicate the study's design with a commonly used term in the title or the abstract. I would add the notion of data linkage, not to misguide the reader, as this is not a cohort. For example: "In-Utero Exposure to Chikungunya and Child Morbidity-Mortality: A Population-Based Longitudinal Study through data linkage."

Thank you very much for this helpful suggestion. In response, we have revised both the title and the abstract to clearly indicate that this is a population-based longitudinal study conducted through record linkage.

We have also added more information in the Methods section of the abstract.

1(b). Provide in the abstract an informative and balanced summary of what was done and what was found.

Page 2, line 20. "The long-term effect of in utero-exposure is not known." This statement may apply for Brazil and similar countries, but French authors have reported longer term neurodevelopmental outcomes in the follow-up of the CHIMERE cohort study (Sarton R et al, *Viruses* 2025), while, in France, a higher standard-care country, the outcome of in utero-exposed uninfected children was the same that that of unexposed-uninfected children (Fritel X et al, *Emerg Infect Dis* 2010; Gérardin P et al, *PLoS Negl Trop Dis* 2014). Please, revise to propose a more contrasted sentence.

We have rephrased to "Chikungunya is an expanding global threat, and in utero exposure has been linked to neonatal morbidity and neurodevelopmental e

ffects,” (line 22) to clarify that previous research from groups such as the La Reunionese team has made substantial advances in the current neonatal CHIK expertise. More details have been added to the introduction section.

Page 2, line 31. With respect to infant mortality, the interpretation overstates the findings, as it only stands for intrapartum exposures, and we have no background information on maternal-neonatal vertical transmission nor on iatrogenic or suboptimal perinatal care, to link the infant death to the infection directly or indirectly.

We appreciate the reviewer’s thoughtful comment and agree that the original phrasing in the findings section of the abstract may have overstated the implications of our results, particularly given the model’s instability due to the small sample size. To avoid implying a direct causal relationship and to ensure that our interpretation accurately reflects the limitations of the available data, we have revised the sentence accordingly (line 32). However, we would like to note that we do not anticipate that the standard of care would differentially affect our estimates, as we expect it to be similar for both exposed and non-exposed groups.

Introduction

2. Background/rationale

Explain the scientific background and rationale for the investigation being reported.

Background science is somewhat biased toward confirmation and preservation of Brazilian findings.

Page 3, line 43. “Current evidence on the effects of in utero CHIKF exposure on newborns remain inconsistent.” Neonates have long been designated as an at-risk population for severe chikungunya. Please, add some antagonist references to support the statement.

When we stated that the current evidence remains inconsistent, we meant that the existing studies report contradictory findings, but we apologize for the inaccuracies and ambiguity. We now make this point more explicitly in the introduction by citing results from international cohorts, ensuring a neutral tone (lines 44–58). We also briefly summarize what is currently known about neonatal vulnerability to chikungunya, citing abundant evidence from early clinical cohorts, such as the La Réunion cohort.

Page 3, lines 43 to 45. “Most studies do not report associations with adverse birth outcomes; however, in cases of vertical transmission, severe neonatal transmission has been documented.” We also expect some references here. Please quote some of the seminal studies that have first report on these background data. There are also more recent studies in Vietnam or Nigeria that have report on birth outcomes.

Thank you for this helpful suggestion. We have referenced and discussed the findings from Nigeria (Ogwuche J 2023), Grenada (Foeller 2021), Thailand (Laoprasopwattana 2016), and French Guiana (Basurko 2022) in addition to the La Reunionese evidence in the introduction in lines 50–55.

Page 3, lines 45 to 47. “A systematic review of 266 in utero-exposed reports that vertical transmission occurs in up-to 48.7% of intrapartum exposed cases…” “That is true however the vertical transmission rate is very difficult to assess given the unreliability of umbilical cord blood antibodies, placenta samples, and lack of studies controlling postnatal PCRs in neonates. The risk vertical transmission rate and fetal mortality risk have also been meta-analyzed by north American authors, this should be quoted here along with the seminal study driving these background data to avoid the above mentioned cognitive biases.

We agree that vertical transmission rates reported in the literature are highly variable and should be interpreted with caution due to methodological limitations, including the availability of umbilical cord serology.

To provide a more balanced and methodologically informed background, we have added a new paragraph on vertically transmitted cases, citing the earliest findings from La Reunion and the North American meta-analysis (Contopoulos-Ioannidis 2018) (lines 44–49). This complements the Ferreira et al. (2021) systematic review already cited in the manuscript and helps contextualize the variability in reported transmission rates. We have also clarified that the 48% vertical transmission reported in the systematic review by Ferreira 2021 stems from La Réunionese studies that conducted intensive serological analyses of mother–fetus pairs (Gerardin 2008).

Page 3, lines 49 to 51. “However, more recently, a nationwide study from Brazil involving 6,000 exposed newborns found that second-trimester CHIKF infection was associated with increased risks of low Apgar scores and neonatal death.¹⁰ “That is true that large powered studies are needed to report on

scarce outcomes but the statement is incomplete. The study also reported an association between symptomatic maternal chikungunya and increased risk of preterm birth (Cerqueira-Silva T et al, Nat Comm 2025). The authors should elaborate here on causality discussing perinatal care in the light of Brazilian perinatal indicators and be more sincere and quote explicitly that is a study by their group. There should be no notoriety bias in reporting novel findings on valuable data.

We have expanded the relevant paragraph (lines 57–59) to provide more information on the findings of our previous study and the Brazilian context, and have clearly stated that this was a study conducted by our team to avoid any impression of notoriety bias.

We have also added to the discussion some information about the limits of registry-based studies, including the potential impact of misclassification due to limited laboratory confirmation and the lack of individual-level clinical data, which may contribute to residual confounding (line 170–).

Pages 3 to 4, lines 52 to 54. “Evidence on the long-term consequences of in utero-exposure is scarce. A study from La Réunion has prospectively followed 33 children for two years, revealing higher prevalence of global neurodevelopmental delay (GND) in the exposed...” That is poor review of the literature and inadequate reporting. For sure, since big countries are involved in chikungunya research, it is very difficult to report updates of Reunionese cohorts with longer term outcomes (Sarton R et al, Viruses 2025; Sarton et al, SSR preprint 2025). For sure, the CHIMERE study may seem small in comparison to their study, but this was indeed a cohort study set up during a real time epidemic, and this study did not report on GND prevalence, but on GND cumulative incidence, as it used incidence rate ratios. The authors should provide the readership with the most recent evidence and quote the literature appropriately.

We fully acknowledge the importance and influence of the La Réunion research program in shaping global understanding of the long-term consequences of perinatal and in-utero chikungunya infection. We have now corrected this with a detailed and appropriately credited discussion.

In the revised introduction in lines 62–66, we have expanded the paragraph on long-term neurodevelopmental outcomes to explicitly and thoroughly incorporate the contributions from La Reunion (Gerardin 2014, Sarton 2025), Grenada (Waechter 2020), Brazil (Quintans 2024), Curacao (Ewijk 2021), and Indi

a (Shukla 2021). We now describe in detail the findings from the prospective CHIMERE cohort established during the 2005-2006 La Réunion epidemic, which remains one of the most rigorously designed and informative studies in this field. We have ensured that our manuscript reflects the most current and comprehensive evidence available with a balanced reference to previous studies.

3. Objectives

State specific objectives, including any prespecified hypotheses.

Page 4, lines 57 to 60. Hypotheses are missing.

Thank you for noting that our initial submission did not explicitly state the study hypotheses. We agree that clearly presenting our prespecified hypotheses improves the clarity and methodological transparency of the manuscript. In response, we have now added a dedicated statement of our hypotheses at the end of the Introduction (line 66-).

Methods

4. Study design

Present key elements of study design early in the paper.

Page 4, lines 63 to 65. The study is indeed longitudinal through data linkage of several valuable, well-powered databases, but this is not a real prospective cohort study providing ad hoc fine data susceptible to argue for causality beyond a literature perspective. The data were gathered from several prospective routinely collected surveillance data.

Thank you for this critical comment. To avoid overstating the nature of our design and to ensure conceptual precision, we have revised the manuscript to remove the term '*cohort*' whenever it might imply a purpose-built prospective study. We retain the term only when referring to entities formally named as such—for example, the *CIDACS Birth Cohort* and the *matched-cohort* analytic design used in our sensitivity analyses. We have also adjusted wording in the Methods and Introduction to explicitly describe the study as a population-based longitudinal study constructed through record linkage of routinely collected administrative and surveillance databases.

5. Setting

Describe the setting, locations, and relevant dates, including periods of recruitment, exposure, follow-up, and data collection.

In the revised manuscript, we added a *Study design and Setting* subsection (lines 212-) to clearly describe the study location, the calendar period of births (2015-2018), the exposure window (conception to delivery), and the follow-up period (birth to 3 years or to December 31, 2018). We also specified that all data were obtained through linkage of routinely collected national Brazilian registries and that data linkage/cleaning were completed in 2023.

6. Participants

6 (a). Cohort study—Give the eligibility criteria, and the sources and methods of selection of participants. Describe methods of follow-up.

Thank you for this comment. We have revised the *Study population* section (line 232-) to explicitly state the eligibility criteria (live-born infants ≥ 20 gestational weeks and ≥ 500 g born between 2015-2018 to mothers aged 15-49 registered in CADU), the exclusion criteria, and the sources of participant selection (SINASC linked to CADU). We also added a clear description of the follow-up process, noting that individuals were followed from birth until the first event, death, age 3, or December 31, 2018, through probabilistic linkage to the national hospitalization (SIH) and mortality (SIM) systems.

Page 5, lines 87 to 88. It is written “birth with gestational age less than 20 weeks; birth with weights below 500 gr...”. These are no births, as in perinatal medicine the term birth is reserved for deliveries at gestational age or birthweights compatible with viability. Before 20 weeks or below 500 gr, the authors should prefer late miscarriage or early fetal losses.

Although SINASC occasionally contains records of very small births (which may, in rare cases, survive under intensive care by neonatologists), we agree that these should not be described as “births” and are not appropriate for inclusion. In the revised manuscript, we removed the phrasing referring to “births <20 weeks or <500g” and instead explicitly defined eligibility as live-born infants registered in SINASC with gestational age >20 weeks and birthweight >500g, thereby ensuring terminology consistent with perinatal medicine (line 235). This change clarifies wording only—the analytic population itself did not change.

6 (b). Cohort study—For matched studies, give matching criteria and number of exposed and unexposed.

The authors provide sufficient information about their matching methods but supplementary material 4 should give p values of comparisons between matched and non-patched participants to let the reader know if there could result selection bias from matching.

Thank you for this suggestion. However, p-values are highly dependent on sample size: larger samples tend to produce smaller p-values, even when the observed differences are not meaningful. Interpreting p-values as evidence against the null hypothesis rather than as indicators of meaningful differences between groups can therefore be misleading. For this reason, we have chosen not to report p-values.

Reference: Greenland, S. Valid p-values behave exactly as they should: Some misleading criticisms of p-values and their resolution with s-values. *Am. Stat.* 73, 106-114 (2019).

7. Variables

Clearly define all outcomes, exposures, predictors, potential confounders, and effect modifiers. Give diagnostic criteria, if applicable

Page 6, line 94. The authors should report more information on SINAN exhaustiveness and representativeness, to allow more comprehension.

Thank you for this comment. As the reviewer has pointed out, a limitation of our study is that no studies to date have reliably quantified underreporting of Arboviruses in SINAN. We have added this point to the Limitations section, lines 191-. However, chikungunya, dengue, and Zika are mandatory notifiable diseases in Brazil, with all suspected cases required to be reported and subsequently undergo epidemiological investigation and final classification by the Ministry of Health.

Page 6, lines 107 to 109. The definition of the intrapartum exposure to the infection is spurious. It is based on a very partial narrative review from Brazilian authors, which endorses a lasting CHIKV viremia of 7-10 days, as found in textbooks. This applies in most nonpregnant adults but not for pregnant women. Indeed, it was shown in a collaboration between Reunion island perinatal specialists and Institut Pasteur researchers, that acute symptoms and the infectious viral loads in 61 parturient women were short-lived (up to 5 days) and needed to peak between day-2 and day+2 around delivery to infect the placenta and being able to be transmitted vertically to the fetus (Gérardin P et al, *PLoS Med* 2008). In addition, regular and intense uterine

contractions needed to be observed to define the intra-partum period, (pre-labour and labour). In accordance, we recommend to discuss the possibility of misclassification using an epidemiological definition and the opportunity to reclassify women (who were indeed in the prepartum period) from the intrapartum to the antepartum group.

We appreciate the reviewer's advice. Following your recommendation and the findings of Gérardin et al (PLoS Med 2008), we performed an additional sensitivity analysis using the cutoff proposed by the La Réunion researchers, defining intrapartum exposure as symptom onset occurring between 2 days before and 2 days after delivery. We reclassified women accordingly and re-estimated the associations, with 3,133 pregnancies classified as antepartum and 65 as intrapartum exposures; among the intrapartum group, 26 infants were hospitalized and 3 died. These results were consistent with our primary analyses, and we have added this to the *Sensitivity Analyses* section (supplementary tables 12 and 13).

8. Data sources/ measurement

For each variable of interest, give sources of data and details of methods of assessment (measurement). Describe comparability of assessment methods if there is more than one group

9. Bias

Describe any efforts to address potential sources of bias, especially indication bias for hospitalization, as in most obstetrical guidelines pregnant women with fever should be referred to the hospital or even, treated as inpatient. In the same manner, discuss indication bias for hospitalization the intrapartum period, as a 50% (or 48.7%) absolute risk of vertical transmission rate followed by a 50% risk of severe neonatal infection is a powerful incentive to monitor neonates with alarms in neonatal care units, at least for three days, until blood virologic samples are negative. Indeed, a negative PCR at day+3 is often reassuring in daily pediatric clinical practice and sometimes used to allow the mother back home with her newborn child, without a risk of developing symptoms.

We appreciate the reviewer's important observation regarding potential indication bias for hospitalization, particularly in the context of fever during pregnancy and neonatal monitoring practices related to intrapartum CHIKV exposure. However, in Brazil, there are no formal guidelines recommending h

ospitalization for pregnant women with chikungunya or for newborns with in utero exposure.

Following the reviewer's suggestion, we conducted an additional analysis excluding infants who died in the neonatal period and those born preterm, resulting in a sample of 2,851 exposed and 25,479 unexposed infants. In this restricted sample, the hazard ratio for hospitalization (1.21 [1.115-1.317]) was very similar to that of the main analysis.

After discussion among the coauthors, we concluded that this analysis may introduce survival bias problems, as it selectively excludes more fragile infants from the study population. For this reason, we chose not to include this analysis in the main results, although we acknowledge the validity of the reviewer's concern.

To address the critical point raised by the reviewer, we evaluated how many newborns were hospitalized for observation. We quantified those hospitalized on the day of birth and discharged within 1-3 days. There were 69 individuals hospitalized for one day after birth (6 exposed and 63 non-exposed), and there were 405 individuals hospitalized for three or fewer days after birth (43 exposed and 362 non-exposed). Notably, 32 out of the 69 hospitalized for one day had been discharged due to death. Among 405 who were hospitalized for 3 days, 34 died during the hospital admission, and another 30 died during the follow-up period. This pattern suggests that short hospitalizations in Brazil are unlikely to represent routine observation of well neonates, limiting the feasibility of isolating "monitoring admissions." Nonetheless, we repeated the survival analyses after excluding infants with 1-day or 3-day early-life hospitalizations. For analyses excluding individuals with a 1-day hospitalization, the HR for hospitalization was 1.22 [95% CI: 1.12-1.33] and for death, 1.19 [95% CI: 0.84-1.70].

Because these exploratory exclusions may themselves introduce survival bias, we chose not to include them as formal sensitivity analyses. We have acknowledged and explained the possibility of indication bias in our limitations (line 308-).

10. Study size

Explain how the study size was arrived at, especially why matching 1 infected neonate with 10 unexposed neonates, and more or less?

No an a-priori sample size calculation was performed for this study, as we used registry-based linked data that included all eligible individuals. Because in-utero CHIKF exposure was rare and matching was done on fine strata

(month of conception and RGI), we used up to 10 unexposed births per exposed to ensure sufficient comparators and maximize statistical precision. We added this to the statistical analyses section, starting at line 294.

11. Quantitative variables

Explain how quantitative variables were handled in the analyses. If applicable, describe which groupings were chosen and why

12. Statistical methods

12 (a). Describe all statistical methods, including those used to control confounding.

Page 8, lines 140 to 141. Justify the preference of standard mean differences (SMD) to compare the distribution of covariates between exposed and nonexposed groups. SMD are less intuitive than p values for non-epidemiologist readership and require normality to gain over p values.

Thank you for pointing this out. We initially used SMD instead of p-values obtained from chi-squared test or Mann-Whitney test, because the p-values tend to get extremely small even in the absence of meaningful differences when we have a large sample size (similar to the response in page 8).

12 (d). Cohort study-If applicable, explain how loss to follow-up was addressed

Thank you for raising this point. Because we are using outcomes derived from national registry data, we assumed that loss to follow-up is negligible; we added this point in the outcomes section (line 274). Also, the rare cases where the individual was hospitalized in a private institution or in a foreign country may not be captured; thus, we added this as a limitation (line 174-).

12 (e). Describe any sensitivity analyses.

Page 9, lines 164 to 165. The “intubation period” is not clear for readers non familiar with perinatal dengue infection or perinatal medicine. The authors should refer to the abovementioned paper (Gérardin P et al, PLoS Med 2008) to justify the extension of the intrapartum period to day+2, as in real life the acme of maternal symptoms may be postponed in postpartum.

We appreciate the reviewer’s valuable input on this point. The term intubation period may indeed be unclear to some readers; thus, we have avoided using it. Also, because Professor Gerardin et al have extensively investigated the viability of the viral load in the intrapartum period using clinical samples from the CHIMERE cohort, and have set up the fundamental evidence f

or future research in in-utero CHIK research, we have rewritten our method section to build our discussion entirely based on this valuable piece of evidence from the La Reunion team. We have referenced their paper and explained our sensitivity analyses in light of it.

Add a fifth sensitivity analysis excluding preterm births and neonatal death from intrapartum transmission to control indication bias and better assess the risk of hospitalization.

Thank you for this insightful suggestion. We agree that our current analyses are prone to indication bias, and additional analyses addressing this point are desirable. However, due to potential bias in these suggested analyses, as detailed on page 8, we have decided not to include them in the manuscript. We have performed the suggested sensitivity analyses, and the results are shown in the answers to the reviewer's comments on page 10.

Results

13. Participants

14. Descriptive data

14 (b). Indicate number of participants with missing data for each variable of interest

Thank you for pointing this out. Because we did a complete case analysis before matching, we did not include the number of participants with missing data in our Table 1. We have included the count of the missingness before the matching in Supplementary Table 2.

14 (c). Cohort study—Summarise follow-up time (eg, average and total amount)

Table 1. All SMD values are small (< 0.5) except that of year of conception that is within the medium range (0.5 to 0.8). The authors should provide additional information on associated p values, for example * $p < 0.01-0.05$ ** $p < 0.001-0.01$ *** $p < 0.001$ to help interpret SMD values.

As explained before, we have not included p-values because they tend to become extremely small even in the absence of meaningful differences when the sample size is large. However, we have calculated the values and have attached Table 1 with the p-values at the end of this document (pages 27-28).

Page 10, lines 182 to 184. “The exposed group had a higher percentage of individuals residing received adequate antenatal care (83% versus 78%, SMD

0.136)…” These findings are counter intuitive. We expect more impoverished people in the exposed group, who are gathered in urban favelas, but it is surprising that these people have more antenatal care. This could reflect both an indication bias and an information bias, those having more antenatal care visits including additional visits due to exposure to the CHIKV. This could also reflect Brazilian health promotion programs before 2019 directed towards reduction of social inequalities in health.

Thank you for this insightful comment. In our setting, although public healthcare is free, access remains unequal due to factors such as distance to facilities, service quality, and waiting times. The higher proportion of antenatal care visits among exposed women may be partly explained by residential location, as a larger proportion of exposed individuals live in urban areas with greater access to healthcare. We have adjusted for both variables in our model.

We also appreciate the reviewer’s suggestion that some women may have received additional antenatal visits due to CHIKV-related symptoms, potentially contributing to the indication. We have now incorporated this explanation into the Discussion.

15. Outcome data

15 (a). Cohort study—Report numbers of outcome events or summary measures over time.

Page 11, lines 187 to 189. 687 or 686 (Suppl. Mat 5) ? 3157 or 3154 (Table 1, Suppl Mat 5) ? 5705 or 5619 (Suppl Mat 5) ? 380 or 379 (Suppl Mat 5).

Supplement material 5: Hospitalizations. The sum of neonates exposed in first, second and third trimester gestation (281+214+215) is over the total number of exposed (686 or 687). Check, revise or explain.

Thank you for pointing this out. I have revised and corrected the numbers.

16. Main results

16 (b). Report category boundaries when continuous variables were categorized

17 Other analyses

Report other analyses done—eg analyses of subgroups and interactions, and sensitivity analyses.

Page 12, line 219. The term “consistent” is a little strong. Check and temper, please.

Thank you for this suggestion. We have tempered the wording by replacing th

e term “consistent” with a more neutral description. The sentence has been revised to state that the sensitivity analyses “replicated the main analyses’ results for hospitalization” (line 111-120).

Page 13, line 224. The data are not shown. Please, check and provide the data in supplement material or add “(data not shown)” .

Thank you for pointing this out. The data referred to in this sentence are provided in Supplementary Material 13; however, we realized this was not clearly indicated in the main text. We have now added an explicit reference to Supplementary Table 13 in the manuscript to ensure readers can easily locate the data (line 112).

Discussion

18. Key results

Summarise key results with reference to study objectives

Page 13, line 233. “Our study is the first to measure health outcomes with a three-year follow-up period.” Contains an unnecessary a priority claim (i.e., This the first study to assess, or we are the first to describe). Instead of generating interest in a study, this claim tends to trivialize the findings. In addition, such claims may offend authors whose earlier papers on the topic may have appeared elsewhere. We strongly suggest that you revise the phrase to include the word “novel” or “unique” even though Reunione studies have similar long and even longer hindsight.

We thank the reviewer for this valuable comment. We agree that our original phrasing could unintentionally be interpreted as an overstatement and may risk diminishing the substantial contributions from previous research. We have revised the sentence to avoid any priority claim. The revised text now emphasizes that our study provides a unique contribution from an LMIC context rather than being “the first,” and explicitly acknowledges the extensive long-term evidence generated by cohorts from other countries (line 122-).

Page 13, line 238. “A study from Brazil…” is to be replaced by “A previous study by our group.” Which is more appropriate.

Thank you for this suggestion. Following the reviewer’s earlier guidance regarding notoriety bias, we have revised the phrase “A study from Brazil …” to “A previous study by our group” to more accurately and transparently acknowledge our prior work (line 138).

19. Limitations

Discuss limitations of the study, taking into account sources of potential bias or imprecision. Discuss both direction and magnitude of any potential bias, especially indication bias for hospitalizations in the intra partum period (see above). Also, discuss this bias in the light of the fifth requested sensitivity analysis.

Thank you for highlighting this point. In the revised manuscript, we have expanded the *Limitations* section to explicitly discuss the potential for indication bias in hospitalization, particularly in the intrapartum period, when neonates might be admitted for observation. As explained above, although we explored several analytic approaches to address this bias, we opted not to include them in the paper as they were subject to survival bias. Still, we now clearly acknowledge in the Discussion how this bias could affect the results.

20. Interpretation

Give a cautious overall interpretation of results considering objectives, limitations, multiplicity of analyses, results from similar studies, and other relevant evidence

Page 13, line 241. The authors assimilate hospitalizations automatically as complications but admissions to the hospital may also result from an indication bias (need for close monitoring of healthy neonates at-risk as born with a mother positive for CHIKV in the intrapartum period).

Thank you for this important observation. We have revised the Interpretation section to avoid suggesting that all hospitalizations necessarily represent clinical complications. We now explicitly acknowledge that some admissions may reflect indication bias, such as precautionary monitoring rather than true morbidity. In addition, we strengthened this section by explicitly referencing and contextualizing our findings alongside previous works, to ensure a more cautious, balanced, and well-grounded interpretation of the results.

21. Generalisability

Discuss the generalisability (external validity) of the study results

Page 14, line 257. “Evidencing that CHIKF can cause serious harms to the fetus highlights the need for preventive measures…” Causal inference cannot be claimed from this study, only presumptions.

We agree that causal inference cannot be drawn from this study. In the revised manuscript, we removed wording that implied causality—such as “can cau

se serious harms” –and replaced it with language that reflects associations and presumptive risks.

Additional comments

Ref#10 lacks year of publication; issue (volume) and pagination.

Thank you for pointing this out. I have revised the reference.

Reviewer #2 (Remarks to the Author):

This is a well-designed and policy-relevant study that addresses an important evidence gap regarding the long-term consequences of in-utero Chikungunya virus (CHIKV) exposure on child morbidity and mortality. The use of a large, population-based cohort, robust data linkage, and matched design makes this work both timely and valuable for public health audiences in endemic and emerging CHIKV regions. The manuscript is generally well written and methodologically sound. My comments below focus on clarifying methodological details, improving transparency, and strengthening the interpretation of results and discussion of limitations.

We are very grateful for your careful reading of the manuscript and your thoughtful, constructive feedback. We particularly appreciate your recognition of the policy relevance of documenting long-term consequences of in-utero CHIKV exposure, as well as your positive assessment of the study design. Following your suggestions, we have clarified several methodological aspects (the definition of in-utero exposure, the rationale for the follow-up window and outcomes, the choice of time scale and risk measures, and the handling of missing data), reorganized the description of the sensitivity analyses, and refined the labelling and presentation of results. We have also strengthened the Discussion by more clearly acknowledging limitations as mentioned by the reviewer. We believe these revisions have substantially improved the manuscript.

Introduction

The Introduction is logically structured and timely, highlighting the public health relevance of Chikungunya virus (CHIKV) and the lack of longitudinal evidence on maternal infection during pregnancy. It can be improved by the following minor revision:

- Line 44: Early clarification of the definition of in-utero CHIKV exposure (i.e., maternal CHIKV infection during pregnancy), this is vital to align the term with the Methods and avoid confusion.

Thank you for highlighting this point. We agree that the term *in-utero CHIKV exposure* should be clearly defined early in the manuscript. In response, we have added an explicit definition in the Introduction (line 45) stating that in-utero exposure refers to *maternal CHIKV infection at any time during pregnancy*, operationalized using SINAN records.

- Line 44: Typo: “remains inconsistent.”

This phrase has been removed after revision.

- Line 59: The Introduction should explain briefly why hospitalization and death up to age three were chosen to represent “long-term morbidity and mortality.”

Thank you for this helpful suggestion. In response, we have added an explanatory sentence on lines 67–69 regarding the choice of our outcome variable. The revised text clarifies that we selected all-cause hospitalization and mortality in the first three years of life as global indicators of long-term health burden, as this composite outcome captures overall morbidity patterns during a critical developmental period and provides a foundation for future cause-specific analyses.

Methods

The Method is generally comprehensive and clearly described, using robust national datasets and appropriate linkage procedures. The matched cohort design is well justified. However, several points need clarification or refinement to strengthen transparency and reproducibility:

- Study population:

Lines 86-90 - Depending on the electronic health records, have you considered the continuity of registration before conception (for data completeness) or deregistration of infants or young children due to migration (for censoring)?

Regarding continuity of registration before conception, because CIDACS performs linkage across national administrative systems, which are not hospital-specific medical records but nationwide registries of births and deaths, the lack of registration is unlikely. In addition, SINASC has high national coverage, with more than 94% of all births registered, further reducing the chance of missing data. However, if a mother did not seek care or were clinically misclassified (misdiagnosed as flu when she actually had CHIK), she would not appear in SINAN as a CHIK case. This may introduce misclassification bias. We have now added this point in our limitations, line 194-.

Regarding censoring, deaths are comprehensively captured in the national mortality registry (SIM), so loss to follow-up due to internal migration is u

unlikely. External migration can lead to loss to follow-up, but is expected to be minimal. The percentage of external migration is approximately 4% in the general population and likely lower among children of low-income families registered in the CADU system (line 175-).

Hospitalizations occurring in private facilities or outside Brazil are not recorded in SIH and may therefore result in under-ascertainment of hospitalization outcomes. Given that our eligible population is drawn from CADU and represents the poorer half of the population, hospitalization in private facilities is expected to be uncommon, although still possible. We have now explicitly added this point to the Limitations section, line 172-.

· Exclusion criteria and exposure classification:

Lines 101-103 - The exclusion of women with >30-day gaps between CHIKF notifications should be supported by references confirming that reinfection within this interval is biologically implausible.

Thank you for this thoughtful comment. We agree that the rationale for excluding women with >30-day gaps between CHIKF notifications should be explicitly supported by biological evidence. In the revised manuscript, we have added that current evidence indicates that CHIKV infection typically induces long-lasting, likely lifelong immunity, making true reinfection within this timeframe biologically implausible. We reference Nitatpattana 2014 to support this justification (line 252).

Lines 106-108 - Please also clarify the reason for the trimester classification (conception-97 days, 98-195 days, 196 to birth).

In the revised manuscript, we now clarify the rationale for our trimester classification. The cutoffs used in our study (conception-97 days, 98-195 days, and ≥ 196 days until birth) were based on the clinical definitions presented in Williams Obstetrics, 24th edition (Cunningham et al, 2014). We acknowledge that alternative studies have used slightly different gestational age boundaries.

We also recognize that gestational age estimation in administrative datasets relies on the reported date of the last menstrual period, which is subject to menstrual cycle variability and recall bias. Therefore, some degree of misclassification across trimesters is possible, which is a limitation of this study.

Lines 109-111 - Intrapartum exposure: Add a brief explanation for stratifying intrapartum infections, e.g., “to distinguish probable vertical transmission during maternal viremia near delivery from indirect effects of earlier maternal infection.”

Thank you for this helpful suggestion. We have added a brief explanation to clarify the rationale for analyzing intrapartum exposure separately in the methods section (line 257-).

· Study design and follow-up:

The study design clearly describes the follow-up window (until age 3, 31 December 2018, or the outcome event, whichever occurred first). However, the primary outcome was first hospitalization up to age 2 years 11 months and 29 days; rephrasing this for consistency would improve clarity.

Thank you for this comment. The reviewer is correct that our original wording was imprecise. The follow-up period ends on the day before the child’s third birthday. We have revised the text to ensure consistency across the manuscript, using terms “up to age 3 years” or “up to 2 years 11 months and 29 days” .

The rationale for choosing a three-year follow-up should also be briefly explained (e.g., data completeness or early-childhood morbidity relevance).

Thank you for this suggestion. We have now clarified the rationale for selecting a three-year follow-up window in line 159. Although linkage allowed potential follow-up for up to 4 years, only a minority of children—especially in the exposed group—had complete 4-year data. We therefore used a three-year follow-up to ensure consistency and minimize differential follow-up time (line 269-).

· Covariate adjustment and DAG:

The DAG is informative, but as maternal socioeconomic status (SES) appears to cause urbanicity, adjusting for both may be redundant. Once SES is included, the backdoor path through urbanicity is already blocked; further adjustment may reduce precision without addressing additional confounding.

We appreciate the reviewer’s insightful comments. In our DAG, maternal SES is not conceptualized as a cause of urbanicity. Rather, we consider the direction of influence to operate primarily from urbanicity to maternal SES, although not so strongly or exclusively that urbanicity can be treated as an

ancestor of maternal SES. For this reason, we believe it remains important to adjust for both sets of variables. We have revised the DAG to better reflect these considerations.

- Missing data:

Lines 139-141 - Complete-case analysis is acceptable as long as the limitation is acknowledged in the Discussion. However, the overall exclusion proportion (~13.9% of eligible participants) should also be reported. Distinguishing between variable-level and individual-level missingness would improve transparency.

Thank you for this observation. We now explicitly report the overall exclusion rate (~13.9% of eligible participants) and clarify the distinction between individual-level and variable-level missingness in the methods section (line 300). The limitations of complete-case analysis are also acknowledged in the Discussion as recommended (line 181-).

- Time scale and risk measures:

Lines 144-150 - Clarify that months were used as the time scale, meaning hazard ratios represent relative incidence rates per person-month, and that standardized risk differences represent cumulative risks per 1,000 children at 12, 24, and 36 months.

Thank you for this helpful suggestion. We now clarify in the Methods section that the analysis used months as the underlying time scale, meaning that hazard ratios represent relative incidence rates per person-month. We also specify that the standardized risk differences correspond to cumulative risks per 1,000 children at 12, 24, and 36 months. These clarifications were added at lines 304-311.

- Sensitivity analyses:

Sensitivity analyses are currently spread across several sections. Consolidating them into a single “Sensitivity analyses” subsection would improve structure and readability.

Thank you for the suggestion. We have added a Sensitivity analyses section in the methods, starting line 319.

Results

- Table 1 (Outcome labels):

The labels “All-cause hospitalization until age 3” and “All-cause death until age 3” should be revised to reflect that these represent the first h

ospitalization/death during follow-up, with censoring at the third birthday or 31 December 2018.

Thank you again for this comment. We have revised to “up to age 3 years” .

· Table 1 (SMD):

Define “SMD” in the table footnote as “Standardized Mean Difference” for clarity.

We appreciate the reviewer’s detailed review; we have added the footnote.

· Figure 2 (Kaplan-Meier curves):

The x-axis should be labelled “time since birth (months)” to clarify that it represents follow-up time to first hospitalization or death.

Thank you for pointing this out; we added the x-axis label.

· Figures 3 and 4 (Sample size):

The figure titles indicate N = 39,694, but the analytic cohort throughout the manuscript is N = 34,694. This should be corrected for consistency.

Thank you for spotting this; it was a typo, and the correct number is 34,694. We have revised the numbers.

· Figure 3 (Subgroup sample sizes):

The figure would be more informative if the number of participants per subgroup (currently reported in Supplementary Material 5) were incorporated directly into the figure. Presenting these counts alongside each HR would improve interpretability and show the precision of subgroup estimates.

Thank you for this advice; we agree that displaying the numbers for each stratum in Figure 3 improves readability, and we have added them to the figure.

· Results text (Interpretation of findings):

Replace terms such as “statistically significant” or “not significant” with phrasing that reflects the strength of evidence (e.g., “strong/moderate/little evidence of an association”), consistent with Nature’s reporting standards and best epidemiological practice.

Thank you for this important point. We have removed all expressions such as “statistically significant” or “not significant” and replaced them with wording that reflects the degree of the evidence for an association. This revision has been applied consistently throughout the Results and Discussion sections.

· Results text (Absolute risk difference):

Clarify that the absolute risk difference (32.9 per 1,000) represents the cumulative risk difference up to 36 months of follow-up (not the incremental difference between 24-36 months). Wording such as “over the first three years of life” would improve interpretability.

Thank you for this helpful suggestion. We have clarified in the Results section that the absolute risk represents the *cumulative* difference in risk from birth up to 36 months of follow-up. We revised the wording to “over the first three years of life” (line 91) to improve clarity and interpretability.

· Terminology (throughout manuscript):

The study population is repeatedly described as “children,” but given that follow-up extends only to age three, “infants” (or “infants and young children”) would be more accurate.

Thank you for pointing this out, and we agree with the reviewer. We have replaced the terms “children” to “infants”.

Discussion

The Discussion is generally well structured, with logical interpretation of findings and thoughtful consideration of biological mechanisms. Several aspects could be clarified or expanded to strengthen interpretability, transparency, and contextual relevance.

Generalisability

· Population representativeness:

The Discussion should explicitly acknowledge that the CIDACS Birth Cohort represents the poorest half of the Brazilian population and thus findings may not generalise to higher-income or nationally representative populations. Thank you for this important point. We have now explicitly acknowledged in the Discussion (line 191-) that the CIDACS Birth Cohort predominantly represents the poorest half of the Brazilian population. We agree that full generalization to higher-income or nationally representative populations may not be possible. However, because we believe the underlying mechanism is biological, we expect that an increased risk could also be observed in other settings. Nonetheless, we emphasize that further studies in different populations are needed to more firmly establish this relationship.

· Matching design:

The matching strategy by month of conception and Immediate Geographic Region (RGI) improves internal validity but limits external generalisability to births within similar time-place strata. This should be noted when discussing policy or public health implications.

We appreciate this observation. The matching approach enhances internal validity by improving comparability between exposed and unexposed groups, without substantially compromising external validity, as the underlying relationships are expected to hold beyond the matched time-place strata.

Potential sources of bias

· Complete-case analysis:

The Discussion should acknowledge that approximately 13.9% of eligible individuals were excluded due to missing covariate data. If missingness was related to SES, healthcare access, or region, selection bias is possible. Clarifying this would strengthen the transparency of limitations.

Thank you for this important point. We explicitly acknowledge in the Methods (line 300) and the discussion (line 181) that approximately 13.9% of eligible individuals were excluded due to missing covariate data. We also note that if missingness was not MCAR, the results may be biased.

Reviewer #3 (Remarks to the Author):

This manuscript addresses a highly relevant and timely public health issue: the effects of in utero exposure to Chikungunya virus (CHIKV) on child morbidity and mortality. The authors present a well-designed, population-based matched cohort study using a robust national dataset from Brazil. The manuscript is clearly written, adheres to STROBE guidelines, and provides transparent methodology and supplementary data.

The study's strengths include its large sample size, rigorous matching procedures, and comprehensive sensitivity analyses. The findings—particularly the increased risk of hospitalization following intrapartum exposure—are compelling and of global relevance given the expanding geographic range of CHIKV due to climate change.

However, several aspects of the manuscript could benefit from further clarification, additional discussion, and minor revisions to improve interpretability and scientific rigor.

We thank the reviewer for the careful and constructive comments. We have revised the manuscript accordingly to improve clarity, contextualization, and interpretation of the findings. Specifically, we expanded the Introduction to describe the Brazilian socioeconomic context better and clarify the relevance of residual confounding; strengthened the Discussion on live birth bias and underreported early pregnancy loss; clarified the absence of data on other congenital infections; and added a brief explanation of the biological mechanisms that may underlie the higher morbidity observed with intrapartum exposure. We believe these revisions have improved the manuscript and appreciate the reviewer's insightful feedback.

Major Comments:

1) Confounding and Effect-modifiers

The introduction would benefit from a more explicit discussion of the limitations in previous studies, particularly regarding confounding and effect-modifiers. For example, how have prior studies failed to account for socioeconomic status, access to healthcare, or co-infections?

The authors should elaborate on how their matched cohort design mitigates these biases and clarify any residual confounding that may remain (in the methods or discussion).

Thank you for this valuable suggestion. We agree with the reviewer that socioeconomic factors are an important confounder, for both Arbovirus incidence and infant hospitalization are largely influenced by socioeconomic factors. We have now clarified in the discussion how our study design helps to mitigate potential biases (line 157-); however, we also acknowledge that residual confounding may still be present (line 175-).

2) Exclusion Criteria and Bias

The rationale for excluding births with ZIKV and DENV exposure is clear, but the manuscript does not mention other congenital infections (e.g., CMV, toxoplasmosis) or congenital anomalies. These are known risk factors for hospitalization and neonatal mortality. Were these considered or excluded? If not, how might they have influenced the results?

We were unable to obtain information on other congenital infections, such as CMV or toxoplasmosis; we have included this as a limitation (line 189).

3) Live Birth Bias

The authors acknowledge the limitation of excluding stillbirths and miscarriages. This live birth bias is particularly relevant for infections occurring in the first trimester. A more detailed discussion of how this may underestimate the true burden of CHIKV exposure in early pregnancy would strengthen the manuscript (miscarriage, birth defects, early fetal demises, early placental abruption, ...).

We thank the reviewer for raising this important point. We have expanded the discussion to describe in more detail how the exclusion of stillbirths and miscarriages may lead to live birth bias, particularly for infections occurring in early pregnancy. As noted in the revised manuscript (line 288), our dataset does not include stillbirths, and infections in the first and second trimesters may result in miscarriage or fetal demise. We now reference evidence from a systematic review reporting a pooled risk of antepartum fetal death of 1.7% following maternal CHIKV infection, as well as pathological studies demonstrating CHIKV infection in placental tissue from spontaneous abortions. We also highlight that early miscarriages are likely underreported in resource-limited settings, which may further contribute to the underestimation of adverse outcomes.

4) Intrapartum Exposure

The finding that intrapartum exposure is associated with an increase in hospitalization risk is striking. The authors should expand on the biological

plausibility of this result. For instance, could the severity be due to direct vertical transmission during delivery, immature neonatal immune response, lack of maternal IgG transfer, or higher viral loads?

We appreciate the reviewer's insightful comment. We expanded the discussion to further elaborate on the biological plausibility of the elevated morbidity observed for intrapartum exposure. As added in the revised manuscript around line 150, clinical evidence indicates that vertical transmission during the intrapartum period may occur through direct contact between the fetal circulation and maternal blood, particularly when maternal viral load is high, as well as through micro-lesions in the placental membranes induced by uterine contractions. We have cited these works as potential biological mechanisms, stating that our work itself has no proof of vertical transmission (we could only identify intrapartum viremia by the infection day, unlike in clinical studies, where vertical transmission is serologically confirmed).

Minor Comments and Suggestions:

Line 46: Consider rephrasing to “severe neonatal morbidity and long-term sequelae have been documented in cases of vertical transmission.”

Thank you for the suggestion. We have rephrased the section per the reviewer's suggestion (line 47-).

Line 48: Replace “sepsis” with “CHIKF clinical syndrome” to avoid confusion with bacterial sepsis.

We understand the reviewer's point; however, many previous studies use the term “neonatal sepsis” to report neonatal CHIK symptoms, such as rash and fever. We have used the word “sepsis-like symptom” to avoid confusion with bacteremia and to align with the term used in previous studies.

Line 85: Clarify whether congenital infections and birth defects were excluded or adjusted for.

We thank the reviewer for this comment. We have clarified this point in the revised manuscript. As noted in the limitations section (line 189), information on other congenital infections, including TORCH pathogens, was not available and therefore could not be excluded or adjusted for. Birth defects were not adjusted for because they may lie on the causal pathway between in-utero CHIKV exposure and subsequent morbidity outcomes.

Line 267: Expand on the limitation regarding underreporting of early pregnancy events, especially in low-resource settings.

We appreciate the reviewer's suggestion. We have expanded the discussion to more clearly acknowledge the limitations of underreporting of early pregnancy losses. As added in the revised manuscript (line 161), early miscarriages are frequently underreported in routine surveillance systems, particularly in resource-limited settings where access to early antenatal care and diagnostic confirmation is limited. Consequently, early fetal losses following maternal CHIKV infection—especially in the first trimester—are likely under-ascertained, which may lead to an underestimation of the true burden of adverse early-pregnancy outcomes.

Discussion: Add a brief explanation of why intrapartum infections may be more severe (e.g., timing of exposure relative to delivery, lack of maternal antibody transfer).

We thank the reviewer for highlighting this point. In response, we have added a concise explanation in the Discussion regarding why intrapartum CHIKV infections may lead to more severe outcomes. As reflected in the revised text (around line 150), studies from clinical settings suggest that transmission close to delivery may be facilitated by high maternal viremia at the time of labour and by mechanical disruption of the placental membranes during contractions, which can allow direct viral passage to the fetus.

	CHIKF exposure status			SMD	P value
	Overall (N=34,694)	Non-exposed (N=31,540)	Exposed (N=3,154)		
Sex					
Female	16864 (48.6)	15342 (48.6)	1522 (48.3)	0.013	0.685
Male	17824 (51.4)	16193 (51.3)	1631 (51.7)		
missing	6 (0.0)	5 (0.0)	1 (0.0)		
Birth weight (grams)					
mean (SD)	3247.02 (537.40)	3246.13 (536.04)	3255.91 (550.91)	0.018	0.34
501-1000g	138 (0.4)	127 (0.4)	11 (0.3)	0.037	<0.001
1001-1500g	196 (0.6)	179 (0.6)	17 (0.5)		
1501-2500g	2047 (5.9)	1849 (5.9)	198 (6.3)		
2501-4000g	30177 (87.0)	27463 (87.1)	2714 (86.0)		
4001-6000g	2134 (6.2)	1920 (6.1)	214 (6.8)		
missing	2 (0.0)	2 (0.0)	0 (0.0)		
Gestational age at birth, weeks					
mean (SD)	38.70 (2.23)	38.70 (2.24)	38.73 (2.20)	0.014	0.466
20-27	161 (0.5)	147 (0.5)	14 (0.4)	0.041	<0.001
28-31	318 (0.9)	292 (0.9)	26 (0.8)		
32-36	3141 (9.1)	2887 (9.2)	254 (8.1)		
37-46	31074 (89.6)	28214 (89.5)	2860 (90.7)		
Marital status					
Divorced	226 (0.7)	201 (0.6)	25 (0.8)	0.056	<0.001
Married	7508 (21.6)	6883 (21.8)	625 (19.8)		
Single	16180 (46.6)	14684 (46.6)	1496 (47.4)		
Stable union	10484 (30.2)	9507 (30.1)	977 (31.0)		

Widowed	68 (0.2)	59 (0.2)	9 (0.3)		
missing	228 (0.7)	206 (0.7)	22 (0.7)		
Apgar score at 5 minutes					
3 or below	99 (0.3)	94 (0.3)	5 (0.2)		
4 to 6	283 (0.8)	250 (0.8)	33 (1.0)	0.041	<0.001
7 or more	33471 (96.5)	30437 (96.5)	3034 (96.2)		
missing	841 (2.4)	759 (2.4)	82 (2.6)		
Maternal education, years					
No education	231 (0.7)	214 (0.7)	17 (0.5)		
1-3 years	1351 (3.9)	1233 (3.9)	118 (3.7)		
4-7 years	8227 (23.7)	7534 (23.9)	693 (22.0)	0.057	<0.001
8-11 years	22310 (64.3)	20207 (64.1)	2103 (66.7)		
12 or more years	2575 (7.4)	2352 (7.5)	223 (7.1)		
Maternal race					
Yellow	86 (0.2)	78 (0.2)	8 (0.3)		
Black (Preto)	1538 (4.4)	1426 (4.5)	112 (3.6)		
Indigenous	196 (0.6)	187 (0.6)	9 (0.3)	0.082	<0.001
Brown (Parado)	28825 (83.1)	26132 (82.9)	2693 (85.4)		
White	4049 (11.7)	3717 (11.8)	332 (10.5)		
Maternal age, years					
mean (SD)	25.43 (6.41)	25.38 (6.40)	25.98 (6.51)	0.094	<0.001
15 to 19	7073 (20.4)	6486 (20.6)	587 (18.6)		
20 to 34	24091 (69.4)	21864 (69.3)	2227 (70.6)	0.051	<0.001
35 to 49	3530 (10.2)	3190 (10.1)	340 (10.8)		
Number of ANC compared to recommended for gestational age					
Adequate	27105 (78.1)	24494 (77.7)	2611 (82.8)	0.136	<0.001

One time less	3171 (9.1)	2920 (9.3)	251 (8.0)		
Two or more times less	4418 (12.7)	4126 (13.1)	292 (9.3)		
Previous pregnancies					
None	12126 (35.0)	11038 (35.0)	1088 (34.5)	0.011	0.574
One or more	22568 (65.0)	20502 (65.0)	2066 (65.5)		
Urbanicity of residence					
Rural	9808 (28.3)	9157 (29.0)	651 (20.6)	0.195	<0.001
Urban	24886 (71.7)	22383 (71.0)	2503 (79.4)		
Year of conception					
2014	7698 (22.2)	7615 (24.1)	83 (2.6)		
2015	8924 (25.7)	7602 (24.1)	1322 (41.9)		
2016	8052 (23.2)	7055 (22.4)	997 (31.6)	0.755	<0.001
2017	8053 (23.2)	7389 (23.4)	664 (21.1)		
2018	1967 (5.7)	1879 (6.0)	88 (2.8)		
All-cause hospitalization up to age 3					
No	28302 (81.6)	25835 (81.9)	2467 (78.2)	0.093	<0.001
Yes	6305 (18.2)	5619 (17.8)	686 (21.8)		
All-cause death up to age 3					
No	34314 (98.9)	31198 (98.9)	3116 (98.8)	0.011	0.5354
Yes	380 (1.1)	342 (1.1)	38 (1.2)		

REVIEWER COMMENTS

Reviewer #1 (Remarks to the Author):

Manuscript Number: NCOMMS-25-77120R1

Type: Research article

Title: In-Utero Exposure to Chikungunya and Child Morbi-mortality: A Population-Based Longitudinal Study

General Comment

I thank the editor for allowing me to re-assess this important manuscript. The authors have done their best for addressing most of my remarks.

I have no additional claim that could be imperative at this step and I think the paper could be published after give some precisions asked in the Specific comment section.

We sincerely thank the reviewer for the careful re-assessment of the manuscript and for the time and expertise invested in providing detailed and insightful comments. We particularly appreciate the reviewer's thorough engagement with the existing literature and the constructive suggestions that helped refine the framing, interpretation, and precision of our work. We have addressed the requested clarifications in the revised manuscript and believe these changes have strengthened the paper.

We would also like to note that, in the current revision, we refined the definition of our exposed population in direct response to the potential exposure misclassification concern raised by Reviewer 1 and we believe that this refinement has meaningfully strengthened the precision of our exposure definition.

In the revised analysis, the exposed group was restricted to chikungunya cases classified as confirmed following the standardized epidemiological investigation process, rather than including all notifications initially registered as suspected cases in SINAN (originally $N = 3,154$). This surveillance process, conducted in accordance with Ministry of Health guidelines, reclassifies arboviral infection notifications as confirmed, inconclusive, or discarded based on clinical evolution, local epidemiological context, and, when available, laboratory results. This refinement strengthens exposure specificity and reduces the potential for non-differential misclassification associated with the inclusion of suspected cases.

Following this restriction, the number of exposed cases decreased ($N = 1,821$); however, effect estimates and overall conclusions remained materially unchanged but enhance confidence in the validity of the classification approach.

Specific comments:

With respect to infant mortality. I agree that the standard of neonatal care is not different between the exposed and the nonexposed children, yet perinatal chikungunya adds an excess mortality risk, due to escalating obstetric care for acute fetal distress. The risk does not exist in other situations of perinatal asphyxia where obstetric care is better codified and better managed by experience. Reduced iatrogenic effects during the perinatal period in the nonexposed group may affect the risk of subsequent hospitalization

beyond the neonatal period differentially. Inversely, a selective survival bias may affect the exposed group that survived the neonatal period likely affecting hospitalization rates. Indeed, the balance between the two trends is highly unpredictable with still the possibility to affect the estimates without not necessarily skewing them. This requires great caution in interpretations, but no revision at this step of the publication.

Thank you very much for this thoughtful comment. We appreciate this insightful and important perspective and fully agree that these considerations warrant caution in interpretation. Although no additional revisions were required, we have clarified in the Discussion the current absence of clinical guidelines for the management of newborns with *in utero* exposure to chikungunya.

Introduction

2. Background/rationale

“Evidence on the long-term consequences of *in utero* exposure is scarce”

Page 4, lines 9 to 11. I agree with the statement, however the outcomes of long-term consequences are very different according to what is meant by exposure. The exposure is qualitatively very different in the 4 studies cited. The Grenadian study (ref#14) reports on neonates that were exposed antenatally, however none was proven infected (the infected term was used for children with positive CHIKV-specific IgG antibodies which indeed indicates maternal infection with passive transfer of transplacental antibodies; or children with IgG+/IgM+ status that indicated postnatal infection). The Brazilian study (ref#15) reports quasi exclusively on mothers infected antenatally or before conception with only one case exposed perinatally who suffered acute fetal distress but got not infected. This study likely suffers a “Zika background bias.” The Curaçao paper (ref#16) refers to both perinatal (n=5) and postnatal (n=17) infections (this is specified in both the abstract and the text) but the seven cases of cognitive delays are observed in postnatally infected children (cf discussion). The Indian paper (ref #17) refers to 13 children born “extramurally” without providing information on mothers nor on confirmatory diagnosis (no child pcr or CHIKV-specific serology) or the lag between birth and presentation to the hospital. The information can be found in a companion paper by the same team (Maria A, et al, Indian Pediatr 2018) which precises that the infection was acquired through perinatal transmission (n=10), postnatal transmission (n=2) or the timing was unknown (n=1). I would suggest to remove the Curaçao paper and rather to consider “the findings are mixed” to write that evidence is increasingly showing an increased risk of adverse neurodevelopmental outcomes adding the earlier contribution of the CHIMERE cohort study (Gérardin P et al, PLoS Negl Trop Dis 2014) and the last Brazilian contribution by the Rio cohort (Pinho de Almeida Di Maio Ferreira FC et al, J Pediatr 2015). Both provide similar conclusions on early neurodevelopmental outcomes with very similar sample sizes.

Thank you very much for this insightful and important comment. In response, we have revised the Background (lines 58–60) to more precisely define *in utero* exposure, clarify differences in exposure ascertainment and timing across prior studies, and exclude the Curaçao study, which primarily reported postnatal infections.

Page 4, line 11. I would add “Novel” to “Evidence from La Réunion”... Unfortunately our last paper which reports now on 13-15 year outcomes is still in the limbs (Sarton R et al, SSRN preprint 2024),

Thank you very much for this helpful suggestion. We agree that the evidence from La Réunion represents a novel and important contribution to the field, and we have added the term “novel” accordingly in the revised manuscript. We are very interested in the forthcoming long-term follow-up of the CHIMERE cohort and look forward to the publication of these 13–15-year outcome data, which will provide valuable insights into the persistence of neurodevelopmental effects following in-utero and perinatal CHIKV exposure.

3. Objectives

State specific objectives, including any prespecified hypotheses.

Page 4, line 17. I would specify in the hypothesis “beyond the neonatal period” as exposed children may be hospitalized in their first days of life just for surveillance and monitoring the onset of symptoms.

Thank you for the suggestion, and we agree to this reviewer’s recommendation for that is precisely our intention. We have added the phrase “beyond the neonatal period” in the last paragraph of the introduction (line 69).

Methods

6 (b). Cohort study—For matched studies, give matching criteria and number of exposed and unexposed.

Matching methods.

I agree with the idea that p-values are highly dependent on the sample size. I understand that the authors having a large sample size have preferred reported standardized mean differences (SMDs), as routinely done in meta-analyses, large RCTs and large observational studies.

It is true that a SMD < 0.1 is often taken for defining perfect balance, however it has also been shown controversial. Conversely, for several authors there is no strict consensus on which SMD value (>0.1, >0.20 or >0.25) define that matching is unbalanced (Ho DEK et al, Political Analysis 2007; Harder VSE, Psychol Methods 2007; Hedges LV, Educ Psychol Meas 2024; Harden JJ 2025). It is also known that SMD can falsely declare studies as biased when sample size is too small or too large (Austin PC et al, Stat Med 2009). Similarly, there is no consensual method to balance diagnostics after propensity score matching, that means correcting the imbalance between the treated and untreated group (Zhang Z et al, Ann Transl Med 2019).

As the authors provided p-values in their rebuttal latter, I would suggest reporting SMDs in both Table 1 and supplementary material 4 and adding a p-values column in supplemental material 4.

In this document, the columns could be as follows:

Not matched Overall Matched Matched p-values SMDs

Nonexposed Nonexposed exposed

This will suggest that matching is well balanced as SMDs are never large (> 0.8) and that except for three variables (number of ANC, urbanicity of residence, and year of

conception), very significant p-values are often consistent with small SMDs, as a consequence of the large sample size.

Thank you very much. In response, we have revised Supplementary Material 4 to report both standardized mean differences (SMDs) and p-values for the matched exposed versus matched non-exposed groups, while retaining SMDs as the primary balance diagnostic. We believe this revision improves transparency regarding balance assessment in the context of a large sample size.

9. Bias

I acknowledge the authors have provided substantial efforts but I still don't understand why the authors do not wish to provide the demanded sensitivity analyses, especially in preterm birth, or why they might introduce a survival bias, while results seem consistent for hospitalization. Do they mean a selective survival bias? Maybe these elements should be introduced in the Method sections and/or discussion, because this will likely question physicians unfamiliar with Brazilian standard cares in obstetrics and perinatal medicine.

Thank you for this clarification. We agree that preterm birth is a key factor that may influence early hospitalization and, therefore, warrants careful consideration. To address this concern, we conducted sensitivity analyses excluding preterm births and low birthweight infants, which directly target the role of prematurity-related admissions; results were consistent with the primary analyses.

We did not include additional analyses excluding neonatal deaths, as conditioning on survival beyond the neonatal period may introduce selective survival bias and potentially distort associations with subsequent hospitalization. We have added corresponding explanations in the Methods (lines 327–331) and Discussion (lines 184–191), clarifying the potential indication bias of early hospitalization and how the sensitivity analyses partially address this concern. We also note that routine observational hospitalization of asymptomatic CHIK-exposed newborns is unlikely in the Brazilian context, where no official guidelines recommend such practice.

Reviewer #2 (Remarks to the Author):

I thank the authors for their careful and comprehensive responses to my comments. The revisions have substantially improved the clarity, transparency, and interpretation of the manuscript. I have no further concerns and support publication.

We thank the reviewer for the careful evaluation of the revised manuscript and for the supportive comments. We are pleased that the revisions have addressed all concerns.

Reviewer #3 (Remarks to the Author):

The authors responded accurately and appropriately to all of my comments and amended the manuscript accordingly. I have no further comments or suggestions.

We thank the reviewer for the thorough review and for the positive assessment of our responses and revisions

 (R)
 16.1
 Statistics/Data analysis

Copyright 1985-2019 StataCorp LLC
 StataCorp
 4905 Lakeway Drive
 College Station, Texas 77845 USA
 800-STATA-PC

Special Edition

<https://www.stata.com>

979-696-4600 stata@stata.com
 979-696-4601 (fax)

Stata license: Single-user perpetual
 Serial number: 401606346822
 Licensed to: Patrick GÃ©rardin
 CHU Reunion

Notes:

1. Unicode is supported; see help unicode_advice.
2. Maximum number of variables is set to 5,000; see help set_maxvar.
3. New update available; type -update all-

** Table 1 **

. tabi 15342 1522\16193 1631, chi2 expected row col exact

```

+-----+
| Key          |
+-----+
| frequency    |
| expected frequency |
| row percentage |
| column percentage |
+-----+

```

row	col		Total
	1	2	
1	15,342	1,522	16,864
	15,331.1	1,532.9	16,864.0
	90.97	9.03	100.00
	48.65	48.27	48.62
2	16,193	1,631	17,824
	16,203.9	1,620.1	17,824.0
	90.85	9.15	100.00
	51.35	51.73	51.38
Total	31,535	3,153	34,688
	31,535.0	3,153.0	34,688.0
	90.91	9.09	100.00
	100.00	100.00	100.00

Pearson chi2(1) = 0.1650 Pr = 0.685
 Fisher's exact = 0.695

1-sided Fisher's exact = 0.349

. ttesti 31538 3246.13 536.04 3154 3255.91 550.91

Two-sample t test with equal variances

```
-----
---
      |      Obs      Mean      Std. Err.      Std. Dev.      [95% Conf.
Interval]
-----+-----
---
      x |   31,538     3246.13     3.018423     536.04     3240.214
3252.046
      y |    3,154     3255.91     9.809566     550.91     3236.676
3275.144
-----+-----
---
combined |   34,692     3247.019     2.885291     537.4082     3241.364
3252.674
-----+-----
---
      diff |           -9.78     10.03625           -29.45137
9.891368
-----+-----
---

```

```

diff = mean(x) - mean(y)                                t = -
0.9745
Ho: diff = 0                                           degrees of freedom =
34690

Ha: diff < 0                                           Ha: diff != 0                                           Ha: diff > 0
Pr(T < t) = 0.1649                                     Pr(|T| > |t|) = 0.3298                                     Pr(T > t) =
0.8351

```

. tabi 127 11\179 17\1849 198\27463 2714\1920 214, chi2 expected row col exact

```
+-----+
| Key          |
|-----|
| frequency    |
| expected frequency |
| row percentage |
| column percentage |
+-----+
```

```
Enumerating sample-space combinations:
stage 5: enumerations = 1
stage 4: enumerations = 13
stage 3: enumerations = 153
stage 2: enumerations = 4946
stage 1: enumerations = 0
```

row	col		Total
	1	2	
1	127	11	138
	125.5	12.5	138.0
	92.03	7.97	100.00
	0.40	0.35	0.40
2	179	17	196
	178.2	17.8	196.0
	91.33	8.67	100.00
	0.57	0.54	0.56
3	1,849	198	2,047
	1,860.9	186.1	2,047.0
	90.33	9.67	100.00
	5.86	6.28	5.90
4	27,463	2,714	30,177
	27,433.5	2,743.5	30,177.0
	91.01	8.99	100.00
	87.08	86.05	86.99
5	1,920	214	2,134
	1,940.0	194.0	2,134.0
	89.97	10.03	100.00
	6.09	6.79	6.15
Total	31,538	3,154	34,692
	31,538.0	3,154.0	34,692.0
	90.91	9.09	100.00
	100.00	100.00	100.00

Pearson chi2(4) = 3.7027 Pr = 0.448
Fisher's exact = 0.456

. ttesti 31540 38.70 2.24 3154 38.73 2.20

Two-sample t test with equal variances

	Obs	Mean	Std. Err.	Std. Dev.	[95% Conf. Interval]
x	31,540	38.7	.012613	2.24	38.67528 38.72472
y	3,154	38.73	.0391735	2.2	38.65319 38.80681
combined	34,694	38.70273	.0120065	2.236379	38.67919 38.72626

```

---
diff |                -.03    .0417651                -.111861
.051861
-----
diff = mean(x) - mean(y)                                t = -
0.7183
Ho: diff = 0                                           degrees of freedom =
34692

Ha: diff < 0                Ha: diff != 0                Ha: diff > 0
Pr(T < t) = 0.2363        Pr(|T| > |t|) = 0.4726        Pr(T > t) =
0.7637

```

tabi 147 14\292 26\2887 254\28214 2860, chi2 expected row col exact

```

+-----+
| Key          |
|-----|
| frequency    |
| expected frequency |
| row percentage |
| column percentage |
+-----+

```

Enumerating sample-space combinations:

```

stage 4: enumerations = 1
stage 3: enumerations = 15
stage 2: enumerations = 264
stage 1: enumerations = 0

```

row	col		Total
	1	2	
1	147	14	161
	146.4	14.6	161.0
	91.30	8.70	100.00
	0.47	0.44	0.46
2	292	26	318
	289.1	28.9	318.0
	91.82	8.18	100.00
	0.93	0.82	0.92
3	2,887	254	3,141
	2,855.5	285.5	3,141.0
	91.91	8.09	100.00
	9.15	8.05	9.05
4	28,214	2,860	31,074
	28,249.1	2,824.9	31,074.0
	90.80	9.20	100.00
	89.45	90.68	89.57
Total	31,540	3,154	34,694
	31,540.0	3,154.0	34,694.0

	90.91	9.09	100.00
	100.00	100.00	100.00

Pearson chi2(3) = 4.6654 Pr = 0.198
 Fisher's exact = 0.203

. tabi 201 25\6883 625\14684 1496\9507 977\59 9, chi2 expected row col exact

```

+-----+
| Key          |
+-----+
| frequency    |
| expected frequency |
| row percentage  |
| column percentage |
+-----+

```

Enumerating sample-space combinations:

stage 5: enumerations = 1
 stage 4: enumerations = 14
 stage 3: enumerations = 293
 stage 2: enumerations = 26421
 stage 1: enumerations = 0

row	col		Total
	1	2	
1	201	25	226
	205.5	20.5	226.0
	88.94	11.06	100.00
	0.64	0.80	0.66
2	6,883	625	7,508
	6,825.7	682.3	7,508.0
	91.68	8.32	100.00
	21.97	19.96	21.78
3	14,684	1,496	16,180
	14,709.7	1,470.3	16,180.0
	90.75	9.25	100.00
	46.86	47.77	46.94
4	9,507	977	10,484
	9,531.3	952.7	10,484.0
	90.68	9.32	100.00
	30.34	31.19	30.42
5	59	9	68
	61.8	6.2	68.0
	86.76	13.24	100.00
	0.19	0.29	0.20
Total	31,334	3,132	34,466

	31,334.0	3,132.0	34,466.0
	90.91	9.09	100.00
	100.00	100.00	100.00

Pearson chi2(4) = 8.9457 Pr = 0.062
 Fisher's exact = 0.052

```
+-----+
| Key          |
+-----+
| frequency    |
| expected frequency |
| row percentage |
| column percentage |
+-----+
```

Enumerating sample-space combinations:

stage 3: enumerations = 1
 stage 2: enumerations = 11
 stage 1: enumerations = 0

row	col 1	col 2	Total
1	94	5	99
	90.0	9.0	99.0
	94.95	5.05	100.00
	0.31	0.16	0.29
2	250	33	283
	257.3	25.7	283.0
	88.34	11.66	100.00
	0.81	1.07	0.84
3	30,437	3,034	33,471
	30,433.7	3,037.3	33,471.0
	90.94	9.06	100.00
	98.88	98.76	98.87
Total	30,781	3,072	33,853
	30,781.0	3,072.0	33,853.0
	90.93	9.07	100.00
	100.00	100.00	100.00

Pearson chi2(2) = 4.2410 Pr = 0.120
 Fisher's exact = 0.129

. tabi 214 17\1233 118\7534 693\20207 2103\2352 223, chi2 expected row col exact

```
+-----+
```

```

| Key |
|-----|
| frequency |
| expected frequency |
| row percentage |
| column percentage |
+-----+

```

Enumerating sample-space combinations:

```

stage 5: enumerations = 1
stage 4: enumerations = 26
stage 3: enumerations = 1260
stage 2: enumerations = 70425
stage 1: enumerations = 0

```

row	col		Total
	1	2	
1	214	17	231
	210.0	21.0	231.0
	92.64	7.36	100.00
	0.68	0.54	0.67
2	1,233	118	1,351
	1,228.2	122.8	1,351.0
	91.27	8.73	100.00
	3.91	3.74	3.89
3	7,534	693	8,227
	7,479.1	747.9	8,227.0
	91.58	8.42	100.00
	23.89	21.97	23.71
4	20,207	2,103	22,310
	20,281.8	2,028.2	22,310.0
	90.57	9.43	100.00
	64.07	66.68	64.31
5	2,352	223	2,575
	2,340.9	234.1	2,575.0
	91.34	8.66	100.00
	7.46	7.07	7.42
Total	31,540	3,154	34,694
	31,540.0	3,154.0	34,694.0
	90.91	9.09	100.00
	100.00	100.00	100.00

```

Pearson chi2(4) = 9.0944 Pr = 0.059
Fisher's exact = 0.062

```

. tabi 78 8\1426 112\187 9\26132 2693\3717 332, chi2 expected row col exact

```

+-----+
| Key |
+-----+
| frequency |
| expected frequency |
| row percentage |
| column percentage |
+-----+

```

Enumerating sample-space combinations:

```

stage 5: enumerations = 1
stage 4: enumerations = 21
stage 3: enumerations = 565
stage 2: enumerations = 35400
stage 1: enumerations = 0

```

row	col		Total
	1	2	
1	78	8	86
	78.2	7.8	86.0
	90.70	9.30	100.00
	0.25	0.25	0.25
2	1,426	112	1,538
	1,398.2	139.8	1,538.0
	92.72	7.28	100.00
	4.52	3.55	4.43
3	187	9	196
	178.2	17.8	196.0
	95.41	4.59	100.00
	0.59	0.29	0.56
4	26,132	2,693	28,825
	26,204.5	2,620.5	28,825.0
	90.66	9.34	100.00
	82.85	85.38	83.08
5	3,717	332	4,049
	3,680.9	368.1	4,049.0
	91.80	8.20	100.00
	11.79	10.53	11.67
Total	31,540	3,154	34,694
	31,540.0	3,154.0	34,694.0
	90.91	9.09	100.00
	100.00	100.00	100.00

```

Pearson chi2(4) = 16.9951 Pr = 0.002
Fisher's exact = 0.001

```

.

```

. ttesti 31540 25.38 6.40 3154 25.98 6.51

```

Two-sample t test with equal variances

```

-----
---
      |      Obs      Mean   Std. Err.   Std. Dev.   [95% Conf.
Interval]
-----+-----
      x |    31,540     25.38   .036037     6.4     25.30937
25.45063
      y |     3,154     25.98   .1159178    6.51     25.75272
26.20728
-----+-----
combined |    34,694    25.43455   .034426    6.412303    25.36707
25.50202
-----+-----
      diff |           -.6   .1197095           -.8346345   -
.3653655
-----

```

```

---
      diff = mean(x) - mean(y)                                t = -
5.0121                                                         degrees of freedom =
Ho: diff = 0                                                    34692

```

```

      Ha: diff < 0                Ha: diff != 0                Ha: diff > 0
Pr(T < t) = 0.0000              Pr(|T| > |t|) = 0.0000              Pr(T > t) =
1.0000

```

. tabi 6486 587\21864 2227\3190 340, chi2 expected row col exact

```

+-----+
| Key          |
|-----|
| frequency    |
| expected frequency |
| row percentage |
| column percentage |
+-----+

```

Enumerating sample-space combinations:
stage 3: enumerations = 1
stage 2: enumerations = 88
stage 1: enumerations = 0

row	col 1	col 2	Total
1	6,486	587	7,073
	6,430.0	643.0	7,073.0
	91.70	8.30	100.00
	20.56	18.61	20.39
2	21,864	2,227	24,091

	21,900.9	2,190.1	24,091.0
	90.76	9.24	100.00
	69.32	70.61	69.44
3	3,190	340	3,530
	3,209.1	320.9	3,530.0
	90.37	9.63	100.00
	10.11	10.78	10.17
Total	31,540	3,154	34,694
	31,540.0	3,154.0	34,694.0
	90.91	9.09	100.00
	100.00	100.00	100.00

Pearson chi2(2) = 7.2984 Pr = 0.026
 Fisher's exact = 0.025

. csi 587 6486 2227 21864, or

	Exposed	Unexposed	Total
Cases	587	6486	7073
Noncases	2227	21864	24091
Total	2814	28350	31164
Risk	.2085999	.2287831	.2269606
	Point estimate		[95% Conf. Interval]
Risk difference	-.0201832		-.0359715 -.0043949
Risk ratio	.9117801		.8458368 .9828645
Prev. frac. ex.	.0882199		.0171355 .1541632
Prev. frac. pop	.0079659		
Odds ratio	.8885269		.8079753 .9771095

(Cornfield)

chi2(1) = 5.94 Pr>chi2 = 0.0148

. csi 587 6486 340 3190, or

	Exposed	Unexposed	Total
Cases	587	6486	7073
Noncases	340	3190	3530
Total	927	9676	10603
Risk	.6332255	.6703183	.6670754
	Point estimate		[95% Conf. Interval]
Risk difference	-.0370929		-.0694993 -.0046864
Risk ratio	.9446638		.8977422 .9940379
Prev. frac. ex.	.0553362		.0059621 .1022578

```

Prev. frac. pop | .0048379 |
Odds ratio | .8491275 | .7381574 .9767787
(Cornfield)
+-----+
chi2(1) = 5.24 Pr>chi2 = 0.0221

```

```
. tabi 24494 2611\2920 251\4126 292, chi2 expected row col exact
```

```

+-----+
| Key |
|-----|
| frequency |
| expected frequency |
| row percentage |
| column percentage |
+-----+

```

Enumerating sample-space combinations:

```

stage 3: enumerations = 1
stage 2: enumerations = 219
stage 1: enumerations = 0

```

row	col 1	col 2	Total
1	24,494	2,611	27,105
	24,640.9	2,464.1	27,105.0
	90.37	9.63	100.00
	77.66	82.78	78.13
2	2,920	251	3,171
	2,882.7	288.3	3,171.0
	92.08	7.92	100.00
	9.26	7.96	9.14
3	4,126	292	4,418
	4,016.4	401.6	4,418.0
	93.39	6.61	100.00
	13.08	9.26	12.73
Total	31,540	3,154	34,694
	31,540.0	3,154.0	34,694.0
	90.91	9.09	100.00
	100.00	100.00	100.00

```

Pearson chi2(2) = 47.8564 Pr = 0.000
Fisher's exact = 0.000

```

```
. csi 2611 24494 251 2920, or
```

```

| Exposed Unexposed | Total
+-----+

```

Cases	2611	24494	27105
Noncases	251	2920	3171
Total	2862	27414	30276
Risk	.9122991	.8934851	.8952636
	Point estimate		[95% Conf. Interval]
Risk difference	.018814		.0078264 .0298016
Risk ratio	1.021057		1.008805 1.033458
Attr. frac. ex.	.0206226		.0087278 .0323747
Attr. frac. pop	.0019866		
Odds ratio	1.240099		1.083503 1.419325
(Cornfield)			
	chi2(1) =	9.78	Pr>chi2 = 0.0018

. csi 2611 24494 292 4126, or

	Exposed	Unexposed	Total
Cases	2611	24494	27105
Noncases	292	4126	4418
Total	2903	28620	31523
Risk	.8994144	.8558351	.8598484
	Point estimate		[95% Conf. Interval]
Risk difference	.0435793		.0319056 .055253
Risk ratio	1.05092		1.037283 1.064737
Attr. frac. ex.	.048453		.035943 .0608006
Attr. frac. pop	.0046674		
Odds ratio	1.506238		1.328874 1.707273
(Cornfield)			
	chi2(1) =	41.54	Pr>chi2 = 0.0000

. tabi 11038 1088\20052 2066, chi2 expected row col exact

Key
frequency
expected frequency
row percentage
column percentage

| col

row	1	2	Total
1	11,038	1,088	12,126
	11,009.2	1,116.8	12,126.0
	91.03	8.97	100.00
	35.50	34.50	35.41
2	20,052	2,066	22,118
	20,080.8	2,037.2	22,118.0
	90.66	9.34	100.00
	64.50	65.50	64.59
Total	31,090	3,154	34,244
	31,090.0	3,154.0	34,244.0
	90.79	9.21	100.00
	100.00	100.00	100.00

Pearson chi2(1) = 1.2708 Pr = 0.260
 Fisher's exact = 0.265
 1-sided Fisher's exact = 0.134

. tabi 9157 651\22383 2503, chi2 expected row col exact

```
+-----+
| Key |
|-----|
| frequency |
| expected frequency |
| row percentage |
| column percentage |
+-----+
```

row	col		Total
	1	2	
1	9,157	651	9,808
	8,916.4	891.6	9,808.0
	93.36	6.64	100.00
	29.03	20.64	28.27
2	22,383	2,503	24,886
	22,623.6	2,262.4	24,886.0
	89.94	10.06	100.00
	70.97	79.36	71.73
Total	31,540	3,154	34,694
	31,540.0	3,154.0	34,694.0
	90.91	9.09	100.00
	100.00	100.00	100.00

Pearson chi2(1) = 99.5925 Pr = 0.000
 Fisher's exact = 0.000
 1-sided Fisher's exact = 0.000

. csi 2503 22383 651 9157, or

	Exposed	Unexposed	Total	
Cases	2503	22383	24886	
Noncases	651	9157	9808	
Total	3154	31540	34694	
Risk	.7935954	.7096703	.7172998	
	Point estimate		[95% Conf. Interval]	
Risk difference	.0839252		.0689385	.0989118
Risk ratio	1.118259		1.097052	1.139877
Attr. frac. ex.	.1057531		.088466	.1227123
Attr. frac. pop	.0106365			
Odds ratio	1.57295		1.438191	1.720334

(Cornfield)

+-----+
 chi2(1) = 99.59 Pr>chi2 = 0.0000

. tabi 7615 83\7602 1322\7055 997\7389 664\1879 88, chi2 expected row col

+-----+				
Key				

frequency				
expected frequency				
row percentage				
column percentage				
+-----+				
row	col		Total	
	1	2		
1	7,615	83	7,698	
	6,998.2	699.8	7,698.0	
	98.92	1.08	100.00	
	24.14	2.63	22.19	
2	7,602	1,322	8,924	
	8,112.7	811.3	8,924.0	
	85.19	14.81	100.00	
	24.10	41.92	25.72	
3	7,055	997	8,052	
	7,320.0	732.0	8,052.0	
	87.62	12.38	100.00	
	22.37	31.61	23.21	
4	7,389	664	8,053	
	7,320.9	732.1	8,053.0	
	91.75	8.25	100.00	

	23.43	21.05	23.21
5	1,879	88	1,967
	1,788.2	178.8	1,967.0
	95.53	4.47	100.00
	5.96	2.79	5.67
Total	31,540	3,154	34,694
	31,540.0	3,154.0	34,694.0
	90.91	9.09	100.00
	100.00	100.00	100.00

Pearson chi2(4) = 1.1e+03 Pr = 0.000

. tabi 25835 2467\5705 687, chi2 expected row col

```
+-----+
| Key          |
|-----|
| frequency    |
| expected frequency |
| row percentage |
| column percentage |
+-----+
```

row	col		Total
	1	2	
1	25,835	2,467	28,302
	25,729.1	2,572.9	28,302.0
	91.28	8.72	100.00
	81.91	78.22	81.58
2	5,705	687	6,392
	5,810.9	581.1	6,392.0
	89.25	10.75	100.00
	18.09	21.78	18.42
Total	31,540	3,154	34,694
	31,540.0	3,154.0	34,694.0
	90.91	9.09	100.00
	100.00	100.00	100.00

Pearson chi2(1) = 26.0287 Pr = 0.000

. csi 687 2467 5705 25835, or

	Exposed	Unexposed	Total
Cases	687	2467	3154
Noncases	5705	25835	31540
Total	6392	28302	34694

Risk	.1074781	.087167	.0909091	
	Point estimate		[95% Conf. Interval]	
Risk difference	.0203111		.0120377	.0285846
Risk ratio	1.233014		1.138129	1.335809
Attr. frac. ex.	.1889791		.1213653	.2513898
Attr. frac. pop	.0411632			
Odds ratio	1.261074		1.153393	1.378807

(Cornfield)

chi2(1) = 26.03 Pr>chi2 = 0.0000

. tabi 31198 3116\342 38, chi2 expected row col

-----+
Key
frequency
expected frequency
row percentage
column percentage
 +-----+

row	col		Total
	1	2	
1	31,198	3,116	34,314
	31,194.5	3,119.5	34,314.0
	90.92	9.08	100.00
	98.92	98.80	98.90
2	342	38	380
	345.5	34.5	380.0
	90.00	10.00	100.00
	1.08	1.20	1.10
Total	31,540	3,154	34,694
	31,540.0	3,154.0	34,694.0
	90.91	9.09	100.00
	100.00	100.00	100.00

Pearson chi2(1) = 0.3842 Pr = 0.535

Supplementary material 5

** Events in the full population **

** Hospitalizations **

** Overall hospitalizations **

. tabi 25921 5619\2468 686, chi2 expected row col exact

```

+-----+
| Key |
|-----|
| frequency |
| expected frequency |
| row percentage |
| column percentage |
+-----+

```

row	col		Total
	1	2	
1	25,921	5,619	31,540
	25,808.2	5,731.8	31,540.0
	82.18	17.82	100.00
	91.31	89.12	90.91
2	2,468	686	3,154
	2,580.8	573.2	3,154.0
	78.25	21.75	100.00
	8.69	10.88	9.09
Total	28,389	6,305	34,694
	28,389.0	6,305.0	34,694.0
	81.83	18.17	100.00
	100.00	100.00	100.00

Pearson chi2(1) = 29.8513 Pr = 0.000
 Fisher's exact = 0.000
 1-sided Fisher's exact = 0.000

. csi 686 5619 2468 25921, or

	Exposed	Unexposed	Total	
Cases	686	5619	6305	
Noncases	2468	25921	28389	
Total	3154	31540	34694	
Risk	.2175016	.1781547	.1817317	
	Point estimate		[95% Conf. Interval]	
Risk difference	.0393469		.0243427	.054351
Risk ratio	1.220858		1.137966	1.309788
Attr. frac. ex.	.1809038		.121239	.2365175
Attr. frac. pop	.0196828			
Odds ratio	1.282247		1.172645	1.402094

(Cornfield)

```

+-----+
chi2(1) = 29.85 Pr>chi2 = 0.0000

```

** Hospitalizations by trimester (first trimester as ref) **

Impossible to check given wrong numbers of admission (total > 686)

** Hospitalizations in intrapartum vs antepartum (ref) periods **

tabi 2391 649\77 37, chi2 expected row col exact

```

+-----+
| Key |
+-----+
| frequency |
| expected frequency |
| row percentage |
| column percentage |
+-----+

```

row	col		Total
	1	2	
1	2,391	649	3,040
	2,378.8	661.2	3,040.0
	78.65	21.35	100.00
	96.88	94.61	96.39
2	77	37	114
	89.2	24.8	114.0
	67.54	32.46	100.00
	3.12	5.39	3.61
Total	2,468	686	3,154
	2,468.0	686.0	3,154.0
	78.25	21.75	100.00
	100.00	100.00	100.00

```

Pearson chi2(1) = 7.9653 Pr = 0.005
Fisher's exact = 0.007
1-sided Fisher's exact = 0.005

```

. csi 37 649 77 2391, or

	Exposed	Unexposed	Total
Cases	37	649	686
Noncases	77	2391	2468
Total	114	3040	3154
Risk	.3245614	.2134868	.2175016
	Point estimate		[95% Conf. Interval]

```

Risk difference | .1110746 | .0239007 .1982484
Risk ratio | 1.520288 | 1.156545 1.99843
Attr. frac. ex. | .3422297 | .1353559 .4996072
Attr. frac. pop | .0184585 | |
Odds ratio | 1.770296 | 1.187662 2.638948
(Cornfield)

```

```

+-----+
chi2(1) = 7.97 Pr>chi2 = 0.0048

```

** Deaths **

** Overall deaths **

. tabi 31198 342\3116 38, chi2 expected row col exact

```

+-----+
| Key |
|-----|
| frequency |
| expected frequency |
| row percentage |
| column percentage |
+-----+

```

row	col 1	col 2	Total
1	31,198	342	31,540
	31,194.5	345.5	31,540.0
	98.92	1.08	100.00
	90.92	90.00	90.91
2	3,116	38	3,154
	3,119.5	34.5	3,154.0
	98.80	1.20	100.00
	9.08	10.00	9.09
Total	34,314	380	34,694
	34,314.0	380.0	34,694.0
	98.90	1.10	100.00
	100.00	100.00	100.00

```

Pearson chi2(1) = 0.3842 Pr = 0.535
Fisher's exact = 0.530
1-sided Fisher's exact = 0.292

```

. csi 38 342 3116 31198, or

	Exposed	Unexposed	Total
Cases	38	342	380
Noncases	3116	31198	34314
Total	3154	31540	34694
Risk	.0120482	.0108434	.0109529

	Point estimate	[95% Conf. Interval]	
Risk difference	.0012048	-.0027706	.0051802
Risk ratio	1.1111111	.7962982	1.550384
Attr. frac. ex.	.1	-.2558109	.3549984
Attr. frac. pop	.01		
Odds ratio	1.112466	.7953097	1.556113

(Cornfield)

-----+-----

chi2(1) = 0.38 Pr>chi2 = 0.5354

** Deaths by trimester (firs trimester as ref) **

Impossible to check given wrong numbers of admission (total > 3154 and events > 686)

** Deaths in intrapartum vs antepartum (ref) periods **

. tabi 3007 33\91 5, chi2 expected row col exact

```

+-----+
| Key          |
+-----+
| frequency    |
| expected frequency |
| row percentage |
| column percentage |
+-----+

```

row	col		Total
	1	2	
1	3,007	33	3,040
	3,003.2	36.8	3,040.0
	98.91	1.09	100.00
	97.06	86.84	96.94
2	91	5	96
	94.8	1.2	96.0
	94.79	5.21	100.00
	2.94	13.16	3.06
Total	3,098	38	3,136
	3,098.0	38.0	3,136.0
	98.79	1.21	100.00
	100.00	100.00	100.00

Pearson chi2(1) = 13.2142 Pr = 0.000
 Fisher's exact = 0.005
 1-sided Fisher's exact = 0.005

. csi 5 33 91 3007, or exact

	Exposed	Unexposed	Total
-----+-----			

Cases	5	33	38
Noncases	91	3007	3098
Total	96	3040	3136
Risk	.0520833	.0108553	.0121173
	Point estimate	[95% Conf. Interval]	
Risk difference	.0412281	-.0033718	.0858279
Risk ratio	4.79798	1.915187	12.02003
Attr. frac. ex.	.7915789	.4778579	.9168055
Attr. frac. pop	.1041551		
Odds ratio	5.00666	1.974055	12.72649

(Cornfield)

1-sided Fisher's exact P = 0.0055
2-sided Fisher's exact P = 0.0055

Supplementary material 6

** Neonatal events (in comparison to total events and not total numbers) **

** Hospitalizations (in comparison to total hospitalizations and not total numbers **

** Overall neonatal hospitalizations **

. tabi 2520 3099\330 356, chi2 expected row col exact

row	col 1	col 2	Total
1	2,520	3,099	5,619
	2,539.9	3,079.1	5,619.0
	44.85	55.15	100.00
	88.42	89.70	89.12
2	330	356	686
	310.1	375.9	686.0
	48.10	51.90	100.00
	11.58	10.30	10.88
Total	2,850	3,455	6,305
	2,850.0	3,455.0	6,305.0
	45.20	54.80	100.00

| 100.00 100.00 | 100.00

Pearson chi2(1) = 2.6184 Pr = 0.106
 Fisher's exact = 0.113
 1-sided Fisher's exact = 0.057

. csi 356 3099 330 2520, or

	Exposed	Unexposed	Total	
Cases	356	3099	3455	
Noncases	330	2520	2850	
Total	686	5619	6305	
Risk	.5189504	.5515216	.5479778	
	Point estimate		[95% Conf. Interval]	
Risk difference	-.0325712		-.072157	.0070146
Risk ratio	.940943		.8722493	1.015047
Prev. frac. ex.	.059057		-.0150467	.1277507
Prev. frac. pop	.0064255			
Odds ratio	.8772331		.7485663	1.028015

(Cornfield)

chi2(1) = 2.62 Pr>chi2 = 0.1056

** Neonatal hospitalizations by trimester (first trimester as ref) **

. tabi 786 102\1091 148\921 106, chi2 expected row col exact

Key
frequency
expected frequency
row percentage
column percentage

Enumerating sample-space combinations:

stage 3: enumerations = 1
 stage 2: enumerations = 20
 stage 1: enumerations = 0

row	col		Total
	1	2	
1	786	102	888
	787.8	100.2	888.0
	88.51	11.49	100.00
	28.09	28.65	28.15
2	1,091	148	1,239

	1,099.2	139.8	1,239.0
	88.05	11.95	100.00
	38.99	41.57	39.28

3	921	106	1,027
	911.1	115.9	1,027.0
	89.68	10.32	100.00
	32.92	29.78	32.56

Total	2,798	356	3,154
	2,798.0	356.0	3,154.0
	88.71	11.29	100.00
	100.00	100.00	100.00

Pearson chi2(2) = 1.5277 Pr = 0.466
 Fisher's exact = 0.461

.
 . csi 148 102 1091 786, or

	Exposed	Unexposed	Total
Cases	148	102	250
Noncases	1091	786	1877

Total	1239	888	2127
Risk	.1194512	.1148649	.1175364
	Point estimate		[95% Conf. Interval]

Risk difference	.0045863		-.0230893 .0322619
Risk ratio	1.039928		.8204564 1.318108
Attr. frac. ex.	.0383948		-.2188339 .2413367
Attr. frac. pop	.0227297		
Odds ratio	1.045344		.7997343 1.36636

(Cornfield)

+-----
 chi2(1) = 0.10 Pr>chi2 = 0.7460

. csi 106 102 921 786, or

	Exposed	Unexposed	Total
Cases	106	102	208
Noncases	921	786	1707

Total	1027	888	1915
Risk	.1032132	.1148649	.1086162
	Point estimate		[95% Conf. Interval]

Risk difference	-.0116516		-.0396881 .0163848
Risk ratio	.8985623		.695209 1.161398

```

Prev. frac. ex. |           .1014377           |    -.1613979    .304791
Prev. frac. pop |           .0544002           |
Odds ratio |           .8868877           |    .6654022    1.182092
(Cornfield)

```

```

+-----+
chi2(1) =    0.67  Pr>chi2 = 0.4138

```

** Neonatal hospitalizations in intrapartum vs antepartum (ref) periods **

. tabi 316 333\14 23, chi2 expected row col exact

```

+-----+
| Key |
+-----+
| frequency |
| expected frequency |
| row percentage |
| column percentage |
+-----+

```

row	col		Total
	1	2	
1	316	333	649
	312.2	336.8	649.0
	48.69	51.31	100.00
	95.76	93.54	94.61
2	14	23	37
	17.8	19.2	37.0
	37.84	62.16	100.00
	4.24	6.46	5.39
Total	330	356	686
	330.0	356.0	686.0
	48.10	51.90	100.00
	100.00	100.00	100.00

```

Pearson chi2(1) = 1.6514 Pr = 0.199
Fisher's exact = 0.237
1-sided Fisher's exact = 0.132

```

.
.
. csi 23 333 14 316, or

	Exposed	Unexposed	Total
Cases	23	333	356
Noncases	14	316	330
Total	37	649	686
Risk	.6216216	.5130971	.5189504

	Point estimate	[95% Conf. Interval]	
Risk difference	.1085245	-.0524065	.2694556
Risk ratio	1.211509	.9319679	1.574897
Attr. frac. ex.	.174583	-.0729984	.3650379
Attr. frac. pop	.0112792		
Odds ratio	1.558988	.7967752	3.048965

(Cornfield)

 chi2(1) = 1.65 Pr>chi2 = 0.1988

** Deaths (in comparison to total numbers) **

** Overall neonatal deaths **

. tabi 5412 207\660 26, chi2 expected row col exact

+-----+
Key
frequency
expected frequency
row percentage
column percentage
 +-----+

row	col		Total
	1	2	
1	5,412	207	5,619
	5,411.4	207.6	5,619.0
	96.32	3.68	100.00
	89.13	88.84	89.12
2	660	26	686
	660.6	25.4	686.0
	96.21	3.79	100.00
	10.87	11.16	10.88
Total	6,072	233	6,305
	6,072.0	233.0	6,305.0
	96.30	3.70	100.00
	100.00	100.00	100.00

Pearson chi2(1) = 0.0194 Pr = 0.889
 Fisher's exact = 0.831
 1-sided Fisher's exact = 0.477

. csi 26 207 660 5412, or

	Exposed	Unexposed	Total
Cases	26	207	233
Noncases	660	5412	6072

Total	686	5619	6305
Risk	.0379009	.0368393	.0369548
	Point estimate		[95% Conf. Interval]
Risk difference	.0010616		-.014053 .0161762
Risk ratio	1.028816		.6896167 1.534857
Attr. frac. ex.	.0280094		-.4500809 .3484737
Attr. frac. pop	.0031255		
Odds ratio	1.029952		.6815511 1.556558

(Cornfield)

chi2(1) = 0.02 Pr>chi2 = 0.8893

** Neonatal deaths by trimester (first trimester as ref) **

. tabi 884 4\1225 14\1019 8, chi2 expected row col exact

Key	frequency	expected frequency	row percentage	column percentage
-----	-----------	--------------------	----------------	-------------------

Enumerating sample-space combinations:

stage 3: enumerations = 1

stage 2: enumerations = 8

stage 1: enumerations = 0

row	col 1	col 2	Total
1	884	4	888
	880.7	7.3	888.0
	99.55	0.45	100.00
	28.26	15.38	28.15
2	1,225	14	1,239
	1,228.8	10.2	1,239.0
	98.87	1.13	100.00
	39.16	53.85	39.28
3	1,019	8	1,027
	1,018.5	8.5	1,027.0
	99.22	0.78	100.00
	32.58	30.77	32.56
Total	3,128	26	3,154
	3,128.0	26.0	3,154.0

	99.18	0.82		100.00
	100.00	100.00		100.00

Pearson chi2(2) = 2.9596 Pr = 0.228
 Fisher's exact = 0.231

. csi 8 4 1019 880, or exact

	Exposed	Unexposed	Total	
-----+-----+-----+-----				
Cases	8	4	12	
Noncases	1019	880	1899	
-----+-----+-----+-----				
Total	1027	884	1911	
Risk	.0077897	.0045249	.0062794	
	Point estimate		[95% Conf. Interval]	
-----+-----+-----+-----				
Risk difference	.0032648		-.0036983	.0102278
Risk ratio	1.721519		.520145	5.697695
Attr. frac. ex.	.4191176		-.9225407	.8244904
Attr. frac. pop	.2794118			
Odds ratio	1.727184		.5514311	5.405863
(Cornfield)				
-----+-----+-----+-----				
			1-sided Fisher's exact P = 0.2734	
			2-sided Fisher's exact P = 0.4023	

. csi 8 14 1019 1225, or exact

	Exposed	Unexposed	Total	
-----+-----+-----+-----				
Cases	8	14	22	
Noncases	1019	1225	2244	
-----+-----+-----+-----				
Total	1027	1239	2266	
Risk	.0077897	.0112994	.0097087	
	Point estimate		[95% Conf. Interval]	
-----+-----+-----+-----				
Risk difference	-.0035098		-.0114814	.0044619
Risk ratio	.6893866		.2903501	1.63683
Prev. frac. ex.	.3106134		-.6368301	.7096499
Prev. frac. pop	.1407767			
Odds ratio	.686948		.2943372	1.603602
(Cornfield)				
-----+-----+-----+-----				
			1-sided Fisher's exact P = 0.2652	
			2-sided Fisher's exact P = 0.5197	

** Deaths in intrapartum vs antepartum (ref) periods **

. tabi 627 22\33 4, chi2 expected row col exact

```

+-----+
| Key |
+-----+
| frequency |
| expected frequency |
| row percentage |
| column percentage |
+-----+

```

row	col		Total
	1	2	
1	627	22	649
	624.4	24.6	649.0
	96.61	3.39	100.00
	95.00	84.62	94.61
2	33	4	37
	35.6	1.4	37.0
	89.19	10.81	100.00
	5.00	15.38	5.39
Total	660	26	686
	660.0	26.0	686.0
	96.21	3.79	100.00
	100.00	100.00	100.00

Pearson chi2(1) = 5.2866 Pr = 0.021
 Fisher's exact = 0.046
 1-sided Fisher's exact = 0.046

. csi 4 33 22 627, or exact

	Exposed	Unexposed	Total
Cases	4	33	37
Noncases	22	627	649
Total	26	660	686
Risk	.1538462	.05	.0539359
	Point estimate		[95% Conf. Interval]
Risk difference	.1038462		-.035832 .2435243
Risk ratio	3.076923		1.177149 8.042701
Attr. frac. ex.	.675		.1504897 .8756637

```
Attr. frac. pop | .072973 |
Odds ratio | 3.454545 | 1.181875 10.17493
(Cornfield)
```

```
+-----+
1-sided Fisher's exact P = 0.0456
2-sided Fisher's exact P = 0.0456
```

** Deaths (in comparison to total deaths) **

** Overall neonatal deaths **

. tabi 135 207\12 26, chi2 expected row col exact

```
+-----+
| Key |
|-----|
| frequency |
| expected frequency |
| row percentage |
| column percentage |
+-----+
```

row	col 1	col 2	Total
1	135	207	342
	132.3	209.7	342.0
	39.47	60.53	100.00
	91.84	88.84	90.00
2	12	26	38
	14.7	23.3	38.0
	31.58	68.42	100.00
	8.16	11.16	10.00
Total	147	233	380
	147.0	233.0	380.0
	38.68	61.32	100.00
	100.00	100.00	100.00

```
Pearson chi2(1) = 0.8987 Pr = 0.343
Fisher's exact = 0.384
1-sided Fisher's exact = 0.221
```

. csi 26 207 12 135, or

	Exposed	Unexposed	Total
Cases	26	207	233
Noncases	12	135	147
Total	38	342	380

Risk	.6842105	.6052632	.6131579	
	Point estimate		[95% Conf. Interval]	
Risk difference	.0789474		-.0776604	.2355552
Risk ratio	1.130435		.8960673	1.426101
Attr. frac. ex.	.1153846		-.1159876	.2987876
Attr. frac. pop	.0128755			
Odds ratio	1.413043		.6971106	2.860876

(Cornfield)

chi2(1) = 0.90 Pr>chi2 = 0.3431

** Neonatal deaths by trimester (first trimester as ref) **

. tabi 5 4\4 14\10 8, chi2 expected row col exact

Key	frequency	expected frequency	row percentage	column percentage

Enumerating sample-space combinations:

stage 3: enumerations = 1
stage 2: enumerations = 6
stage 1: enumerations = 0

row	col		Total
	1	2	
1	5	4	9
	3.8	5.2	9.0
	55.56	44.44	100.00
	26.32	15.38	20.00
2	4	14	18
	7.6	10.4	18.0
	22.22	77.78	100.00
	21.05	53.85	40.00
3	10	8	18
	7.6	10.4	18.0
	55.56	44.44	100.00
	52.63	30.77	40.00
Total	19	26	45
	19.0	26.0	45.0
	42.22	57.78	100.00
	100.00	100.00	100.00

Pearson chi2(2) = 4.9190 Pr = 0.085
 Fisher's exact = 0.087

. csi 14 4 4 5, or exact

	Exposed	Unexposed	Total	
Cases	14	4	18	
Noncases	4	5	9	
Total	18	9	27	
Risk	.7777778	.4444444	.6666667	
	Point estimate		[95% Conf. Interval]	
Risk difference	.3333333		-.0438619	.7105286
Risk ratio	1.75		.809426	3.783545
Attr. frac. ex.	.4285714		-.2354434	.7356976
Attr. frac. pop	.3333333			
Odds ratio	4.375		.8355198	23.20801
(Cornfield)				
	1-sided Fisher's exact P = 0.0981			
	2-sided Fisher's exact P = 0.1085			

. csi 8 10 4 5, or exact

	Exposed	Unexposed	Total	
Cases	8	10	18	
Noncases	4	5	9	
Total	12	15	27	
Risk	.6666667	.6666667	.6666667	
	Point estimate		[95% Conf. Interval]	
Risk difference	0		-.3578388	.3578388
Risk ratio	1		.5846404	1.710453
Attr. frac. ex.	0		-.710453	.4153596
Attr. frac. pop	0			
Odds ratio	1		.2105919	4.694239
(Cornfield)				
	1-sided Fisher's exact P = 0.6569			
	2-sided Fisher's exact P = 1.0000			

** Deaths in intrapartum vs antepartum (ref) periods **

. tabi 11 22\1 4, chi2 expected row col exact

+-----+
 | Key |

```

|-----|
| frequency |
| expected frequency |
| row percentage |
| column percentage |
+-----+

```

row	col		Total
	1	2	
1	11	22	33
	10.4	22.6	33.0
	33.33	66.67	100.00
	91.67	84.62	86.84
2	1	4	5
	1.6	3.4	5.0
	20.00	80.00	100.00
	8.33	15.38	13.16
Total	12	26	38
	12.0	26.0	38.0
	31.58	68.42	100.00
	100.00	100.00	100.00

```

Pearson chi2(1) = 0.3573 Pr = 0.550
Fisher's exact = 1.000
1-sided Fisher's exact = 0.488

```

```
. csi 4 22 1 11, or exact
```

	Exposed	Unexposed	Total	
Cases	4	22	26	
Noncases	1	11	12	
Total	5	33	38	
Risk	.8	.6666667	.6842105	
	Point estimate		[95% Conf. Interval]	
Risk difference	.1333333		-.2524063	.519073
Risk ratio	1.2		.7276354	1.979013
Attr. frac. ex.	.1666667		-.3743147	.4946977
Attr. frac. pop	.025641			
Odds ratio	2		.2557302	.

```
(Cornfield)
```

```

1-sided Fisher's exact P = 0.4885
2-sided Fisher's exact P = 1.0000

```

```
. csi 4 22 1 11, or exact woolf
```

	Exposed	Unexposed	Total
--	---------	-----------	-------

Cases		4	22		26
Noncases		1	11		12

Total		5	33		38
Risk		.8	.6666667		.6842105
		Point estimate			[95% Conf. Interval]

Risk difference		.1333333			-.2524063 .519073
Risk ratio		1.2			.7276354 1.979013
Attr. frac. ex.		.1666667			-.3743147 .4946977
Attr. frac. pop		.025641			
Odds ratio		2			.1989719 20.10334 (Woolf)

		1-sided Fisher's exact P = 0.4885			
		2-sided Fisher's exact P = 1.0000